# Hybridization led to a rewired pluripotency network in the allotetraploid *Xenopus laevis*

Wesley A Phelps, Matthew D Hurton, Taylor N Ayers, Anne E Carlson, Joel C Rosenbaum, Miler T Lee*

Department of Biological Sciences, University of Pittsburgh, Pittsburgh, United States

**Abstract** After fertilization, maternally contributed factors to the egg initiate the transition to pluripotency to give rise to embryonic stem cells, in large part by activating de novo transcription from the embryonic genome. Diverse mechanisms coordinate this transition across animals, suggesting that pervasive regulatory remodeling has shaped the earliest stages of development. Here, we show that maternal homologs of mammalian pluripotency reprogramming factors OCT4 and SOX2 divergently activate the two subgenomes of *Xenopus laevis*, an allotetraploid that arose from hybridization of two diploid species ~18 million years ago. Although most genes have been retained as two homeologous copies, we find that a majority of them undergo asymmetric activation in the early embryo. Chromatin accessibility profiling and CUT&RUN for modified histones and transcription factor binding reveal extensive differences in predicted enhancer architecture between the subgenomes, which likely arose through genomic disruptions as a consequence of allotetraploidy. However, comparison with diploid *X. tropicalis* and zebrafish shows broad conservation of embryonic gene expression levels when divergent homeolog contributions are combined, implying strong selection to maintain dosage in the core vertebrate pluripotency transcriptional program, amid genomic instability following hybridization.

*For correspondence:
miler@pitt.edu

Competing interest: The authors declare that no competing interests exist.

## Editor's evaluation

This paper reports fundamental findings that substantially advance our understanding of a major research question – how hybridization events influence gene regulatory programs and how evolutionary pressures have shaped these programs in response to such events. This convincing work uses appropriate and validated methodology in line with the current state-of-the-art.

## Introduction

In animals, zygotic genome activation (ZGA) is triggered after an initial period of transcriptional quiescence following fertilization of the egg, during the maternal-to-zygotic transition (MZT; *Lee et al., 2014*; *Vastenhouw et al., 2019*). In mammals, this occurs during the slow first cleavages (*Svoboda, 2018*), a few days removed from the subsequent induction of pluripotent stem cells in the blastocyst by a core network of factors including NANOG, OCT4, and SOX2 (*Li and Belmonte, 2017*; *Takahashi and Yamanaka, 2016*). In contrast, faster-dividing taxa including zebrafish, *Xenopus*, and *Drosophila* activate their genomes in the blastula hours after fertilization (*Foe and Alberts, 1983*; *Kane and Kimmel, 1993*; *Newport and Kirschner, 1982a*; *Vastenhouw et al., 2019*), which leads immediately to pluripotency. In zebrafish, maternally provided homologs of NANOG, OCT4, and SOX2 are required for a large share of genome activation (*Lee et al., 2013*; *Leichsenring et al., 2013*; *Miao*

**Figure 1.** Identifying the first wave of genome activation across the two subgenomes. (**A**) The allotetraploid *X. laevis* genome contains two distinct subgenomes "L" and "S" due to interspecific hybridization of ancestral diploids. (**B**) Triptolide inhibits genome activation, as measured in the late blastula, while cycloheximide inhibits only secondary activation, distinguishing genes directly activated by maternal factors. NF = Nieuwkoop and Faber. (**C**) Heatmap of RNA-seq coverage over exons (left) and introns (right) of activated genes.

The online version of this article includes the following figure supplement(s) for figure 1:

**Figure supplement 1.** Measuring genome activation.

*et al., 2022*); thus, vertebrate embryos deploy conserved pluripotency induction mechanisms at different times during early development.

Beyond vertebrates, unrelated maternal factors direct genome activation and the induction of stem cells, for example Zelda (*Liang et al., 2008*), CLAMP (*Colonnetta et al., 2021*; *Duan et al., 2021*), and GAF (*Gaskill et al., 2021*) in *Drosophila*, although they seem to share many functional aspects with vertebrate pluripotency factors, including pioneering roles in opening repressed embryonic chromatin and establishing activating histone modifications (*Blythe and Wieschaus, 2016*; *Gaskill et al., 2021*; *Hug et al., 2017*). This diversity of strategies implies that the gene network regulating pluripotency has been extensively modified over evolutionary time (*Endo et al., 2020*; *Fernandez-Tresguerres et al., 2010*), though it is unknown when and under what circumstances major modifications arose.

We sought to understand how recent genome upheaval has affected the pluripotency regulatory network in the allotetraploid *Xenopus laevis*, by deciphering how embryonic genome activation is coordinated between its two subgenomes. *X. laevis*'s L (long) and S (short) subgenomes are inherited from each of two distinct species separated by ~34 million years that hybridized ~18 million years ago (*Session et al., 2016*; *Figure 1A*). A subsequent whole-genome duplication restored meiotic pairing. Despite extensive rearrangements and deletions, most genes are still encoded as two copies (homeologs) on parallel, non-inter-recombining chromosomes (*Session et al., 2016*). Previously, homeologs had been challenging to distinguish due to high functional and sequence similarity; however, the recent high-quality *X. laevis* genome assembly has made it feasible to resolve differential expression and regulation genome-wide between the two subgenomes (*Elurbe et al., 2017*; *Session et al., 2016*).

Allopolyploidy often provokes acute effects on gene expression (*Hu and Wendel, 2019*; *Moran et al., 2021*), leading to regulatory shifts over time to reconcile dosage imbalances and incompatibilities between gene copies (*Grover et al., 2012*; *Song et al., 2020*; *Swamy et al., 2021*). This phenomenon has been explored primarily in plants (*Adams and Wendel, 2005*; *Husband et al.,*

*2013*; *Mable, 2004*), but the extent to which this has occurred in the few characterized allopolyploid vertebrates is unclear (*Chen et al., 2019a*; *Li et al., 2021*; *Luo et al., 2020*). For *X. laevis*, there is a broad trend toward balanced homeolog expression across development and adult tissues, with a subtle bias in favor of the L homeolog that emerges after genome activation (*Session et al., 2016*), and an overall ontogenetic and transcriptomic trajectory similar to 48 million-years diverged diploid *X. tropicalis* (*Harland and Grainger, 2011*; *Yanai et al., 2011*). Initial observations in *X. laevis* have demonstrated differential homeologous enhancer activity in the eye (*Ochi et al., 2017*) and a divergent cis-regulatory landscape of histone modifications and recruitment of transcriptional machinery in the early gastrula (*Elurbe et al., 2017*), suggesting that embryonic genome activation is likely also asymmetric between the two subgenomes.

Although *Xenopus* embryos have long been a model for understanding the MZT, for example (*Amodeo et al., 2015*; *Charney et al., 2017*; *Chen et al., 2019b*; *Gentsch et al., 2019*; *Gibeaux et al., 2018*; *Gurdon et al., 1958*; *Kimelman et al., 1987*; *Newport and Kirschner, 1982a*; *Newport and Kirschner, 1982b*; *Paraiso et al., 2019*; *Skirkanich et al., 2011*; *Veenstra et al., 1999*; *Yanai et al., 2011*), ZGA regulators have not previously been identified in *X. laevis*. Here, we identify maternal Pou5f3 and Sox3 as top-level regulators of *X. laevis* pluripotency and ZGA and elucidate the predicted enhancer architecture that differentially recruits them to homeologous gene copies between the two subgenomes. Despite differential subgenome activation, combined transcriptional output converges to proportionally resemble the diploid state, maintaining gene dosage for the embryonic pluripotency program.

## Results

### Identifying divergently activated homeologous genes

At genome activation, the *X. laevis* pluripotency network consists of maternal regulators acting directly on the first embryonic genes (*Figure 1B*). To identify these genes, we performed a total RNA-seq early embryonic time course using our *X. laevis*-specific ribosomal RNA depletion protocol (*Phelps et al., 2021*; *Figure 1A and B*, *Supplementary file 1*). Subtle gene activation is observed in the blastula at Nieuwkoop and Faber (N.F.) stage 8, culminating in 4772 genes with significant activation by the middle of stage 9 (8 hours post fertilization [h.p.f.] at 23 °C) (*Figure 1C*, *Supplementary file 2*). Gene activation was detected through a combination of exon- and intron-overlapping sequencing reads deriving from nascent pre-mRNA (*Lee et al., 2013*) – indeed, two-thirds of these genes had substantial maternal contributions that masked their activation when quantifying exon-overlapping reads alone (*Figure 1C*). These genes fail to be activated in embryos treated at 1 cell (stage 1) with the transcription inhibitor triptolide (*Gibeaux et al., 2018*) when compared to DMSO vehicle control embryos (*Figure 1B and C*, *Figure 1—figure supplement 1*).

To distinguish direct targets of maternal factors (primary activation) (*Figure 1B*), we then performed RNA-seq on stage 9 embryos treated with cycloheximide at stage 8, to inhibit translation of newly synthesized embryonic transcription factors that could regulate secondary activation (*Harvey et al., 2013*; *Lee et al., 2013*). A total of 2662 genes (56% of all activated genes) were still significantly activated in cycloheximide-treated embryos compared to triptolide-treated embryos, representing the first wave of genome activation in the embryo (*Figure 1C*, *Supplementary file 1A*).

We analyzed subgenome of origin for activated genes and found that they are preferentially encoded as two homeologous copies in the genome (p=2.3 × 10$^{-225}$, $\chi$-squared test, 10 d.o.f.; *Figure 2A*). However, a majority of these genes have asymmetric expression between the two homeologs, often with transcription deriving from only the L or S copy alone (*Figure 2B–C*). This asymmetry is more pronounced at stage 8, but balances somewhat as genome activation progresses, suggesting timing differences for homeolog activation that could result from subtle regulatory divergence (*Figure 2A*, *Figure 2—figure supplement 1A–C*); and slightly less pronounced for strictly zygotic genes compared to maternal-zygotic genes (maternal contribution ≥1 TPM; *Figure 2D*, *Figure 2—figure supplement 1D*), which are often reactivated with different homeolog expression patterns compared to their maternal contribution (*Figure 2E*).

After genome activation, a heightened imbalance in favor of the L homeolog emerges, as measured by activation patterns in four differentiated cell lineages (*Johnson et al., 2022*; *Figure 2D*, *Figure 2—figure supplement 1E–I*), that appears to indicate a shift toward more divergent homeolog regulation

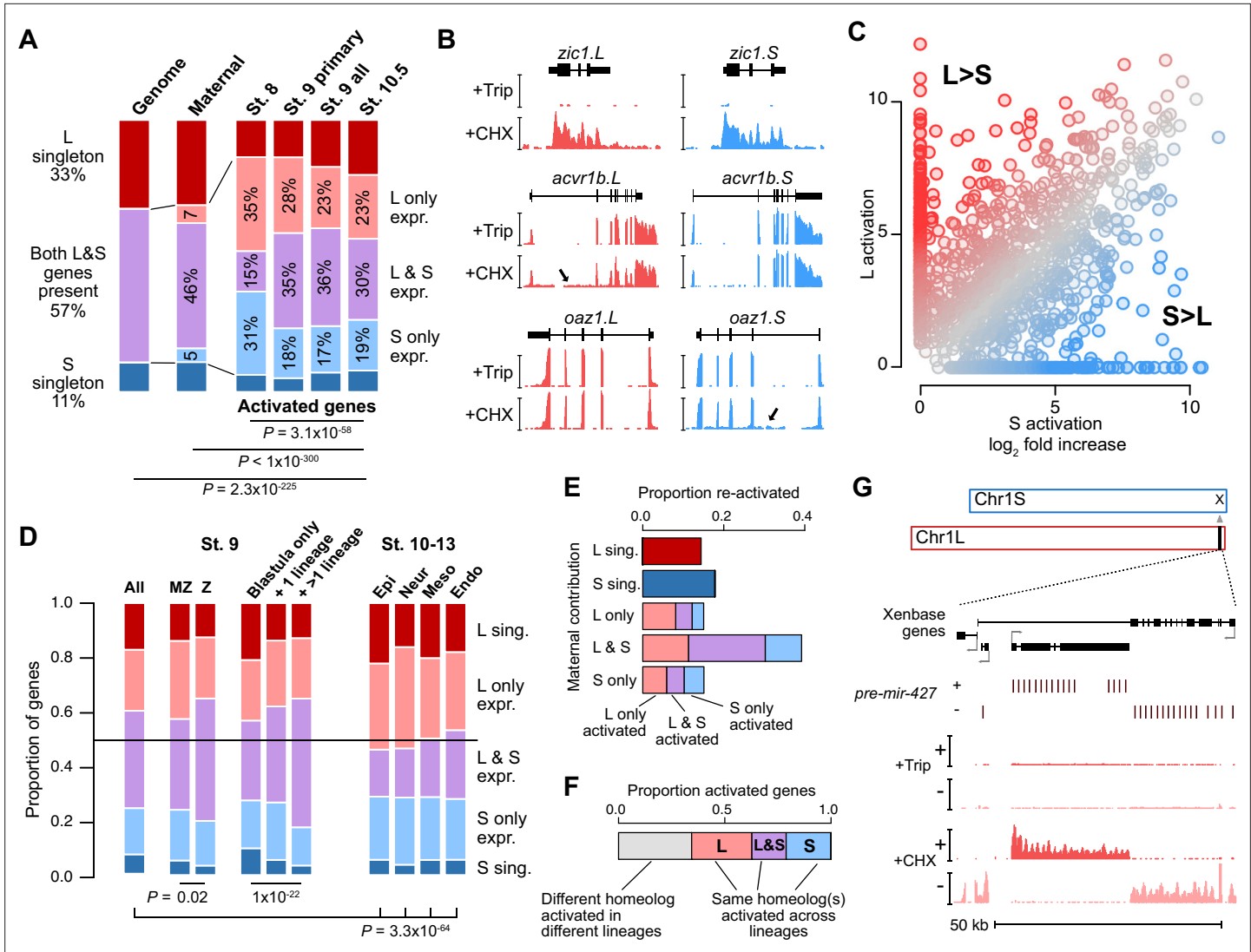

**Figure 2.** Homeologous genes are differentially activated in the early embryo. (**A**) Proportion of genes encoded as homeologs on both subgenomes versus only one subgenome (singleton) (left), as compared to expression patterns in the early embryo. p Values are from $\chi$-squared tests, 10 d.o.f., comparing genomic to expressed proportions, 16 d.o.f., comparing proportions between activated genes and the maternal contribution, 12 d.o.f., comparing proportions at subsequent stages of activation. (**B**) Browser tracks showing log2 reads-per-million RNA-seq coverage of equivalently activated homeologs (top) and differentially activated homeologs (L-specific, middle; S-specific, bottom). Trip = triptolide, CHX = cycloheximide. (**C**) Biplot comparing log2 fold primary activation over triptolide treated embryos for the S homeolog (x axis) versus the L homeolog. (**D**) Left, proportion of genes activated symmetrically or asymmetrically from the L or S subgenomes, stratified into whether there is a maternal contribution for either homeolog (MZ) or not (**Z**) (p=0.02, $\chi$-squared test, 4 d.o.f.); and whether a gene is activated only in the stage 9 blastula or is additionally increased in only one or more than one differentiated lineage from stages 10–13 (p=1.3 × $10^{-22}$, $\chi$-squared test, 8 d.o.f.). Right, homeolog proportions of later gene activation in epidermal (Epi), neural progenitor (Neur), ventral mesodermal (Meso), and endodermal (Endo) lineages from stages 10–13 (p=3.3 × $10^{-64}$, $\chi$-squared test comparing stage 9 and the four lineages, 16 d.o.f.). Lineage-specific gene expression data are from **Johnson et al., 2022**. (**E**) Homeolog-specific stage 9 activation proportions, versus maternal contribution homeolog expression patterns, for maternal-zygotic genes. (**F**) Concordance of homeolog activation patterns across the differentiated lineages at stages 10–13, for genes initially activated at stage 9 and also increased in at least two differentiated lineages. (**G**) Browser track showing strand-separated reads-per-million RNA-seq coverage over the *mir-427* encoding locus on the distal end of Chr1L (v10.1).

The online version of this article includes the following figure supplement(s) for figure 2:

**Figure supplement 1.** Differential homeolog activation over early development.

**Figure supplement 2.** The mir-427 locus.

as development proceeds, as was observed previously (*Session et al., 2016*). However, it is likely that some of the shared homeolog activation as measured in the whole blastula is actually composed of homeolog-specific regional activation (*Chen and Good, 2022*). Indeed, for one-third of genes activated in more than one lineage, different homeologs are activated in different lineages (*Figure 2F*, *Figure 2—figure supplement 1F and G*), and for those genes that are already activated at stage 9, this seems to result in a higher proportion of both-homeolog activation, as compared to genes with single-lineage or blastula-specific activation (p=1.3 × 10$^{-22}$, $\chi$-squared test, 8 d.o.f.). Overall, this indicates a high degree of divergent cis-regulatory architecture between gene homeologs throughout early development.

Genes activated from both subgenomes are enriched in transcriptional regulators (p<0.01, Fisher's exact test, two-sided) (*Figure 2—figure supplement 1J*), suggesting that gene function may have influenced homeolog expression patterns. However, there is no evidence for stronger functional divergence between homeologs expressed asymmetrically between the subgenomes, as estimated by non-synonymous versus synonymous mutation rate in coding regions (dN/dS ratio; *Figure 2—figure supplement 1K and L*).

## The microRNA *mir-427* is encoded on only one subgenome

Among the first-wave genes is the microRNA *mir-427*, which plays a major role in clearance of maternally contributed mRNA (*Lund et al., 2009*). Similar to *X. tropicalis mir-427* (*Owens et al., 2016*) and the related zebrafish *mir-430* (*Lee et al., 2013*), *mir-427* is one of the most strongly activated genes in the *X. laevis* embryonic genome (*Figure 2D*, *Figure 2—figure supplements 1C and 2A*). In version 9.2 of the *X. laevis* genome assembly, the *miR-427* precursor hairpin sequence is found in only five copies overlapping a Xenbase-annotated long non-coding RNA on chr1L (*Figure 2—figure supplement 2B*). To better capture the genomic configuration of the *mir-427* primary transcript, we aligned the miRBase-annotated precursor sequence (*Kozomara et al., 2019*) to the version 10.1 *X. laevis* genome assembly. This revealed an expanded *mir-427* locus at the distal end of Chr1L composed of 33 precursor copies, encoded in both strand orientations over 55 kilobases (*Figure 2D*, *Figure 2—figure supplement 2A and B*). The corresponding region on Chr1S is unalignable (*Figure 2—figure supplement 2C*), suggesting that *mir-427* is encoded on only the L subgenome. We additionally found two *mir-427* hairpin sequence matches to the distal end of Chr3S, but these loci were not supported by substantial RNA-seq coverage (*Figure 2—figure supplement 2D*).

This is reminiscent of the *X. tropicalis mir-427* genomic configuration (*Owens et al., 2016*), although smaller in scale and on a non-homologous chromosome. In *X. tropicalis*, 171 tandemly arrayed *mir-427* precursors are found on the distal end of Chr03, which is thought to accelerate mature *miR-427* accumulation during the MZT to facilitate rapid maternal clearance (*Owens et al., 2016*). Zebrafish similarly encodes a large array of more than 2000 *mir-430* precursors, which begin to target maternal mRNA for clearance shortly after ZGA (*Bazzini et al., 2012*; *Giraldez et al., 2006*; *Hadzhiev et al., 2023*; *Lee et al., 2013*). These results strongly suggest that the mir-427 locus has undergone genomic remodeling, resulting in absence from the S subgenome, but possibly also translocation between chromosomes in the *tropicalis* or *laevis* lineages.

## Subgenomes differ in their regulatory architecture

To discover the maternal regulators of differential homeolog activation, we first profiled embryonic chromatin using Cleavage Under Target & Release Using Nuclease (CUT&RUN) (*Hainer and Fazzio, 2019*; *Skene and Henikoff, 2017*), which we adapted for blastulae. We found that cell dissociation was necessary for efficient nuclear isolation to carry out the on-bead CUT&RUN chemistry (*Figure 3A*, *Figure 3—figure supplement 1A–D*). At stages 8 and 9, the active marks H3 lysine 4 trimethylation (H3K4me3) and H3 lysine 27 acetylation (H3K27ac) were enriched in the transcription start site (TSS) regions of activated genes, and differential homeolog activation measured by RNA-seq significantly correlates with differential histone modification profiles, with a slight overall bias toward stronger L homeolog chromatin activity (*Figure 3B and C*, *Figure 3—figure supplement 1E–G*, *Supplementary file 3*). Differential promoter engagement by transcriptional machinery likely underlies the differential histone modification levels; however, we found no promoter sequence differences between homeologs that would implicate differential recruitment of specific transcription factors (*Supplementary file 4*).

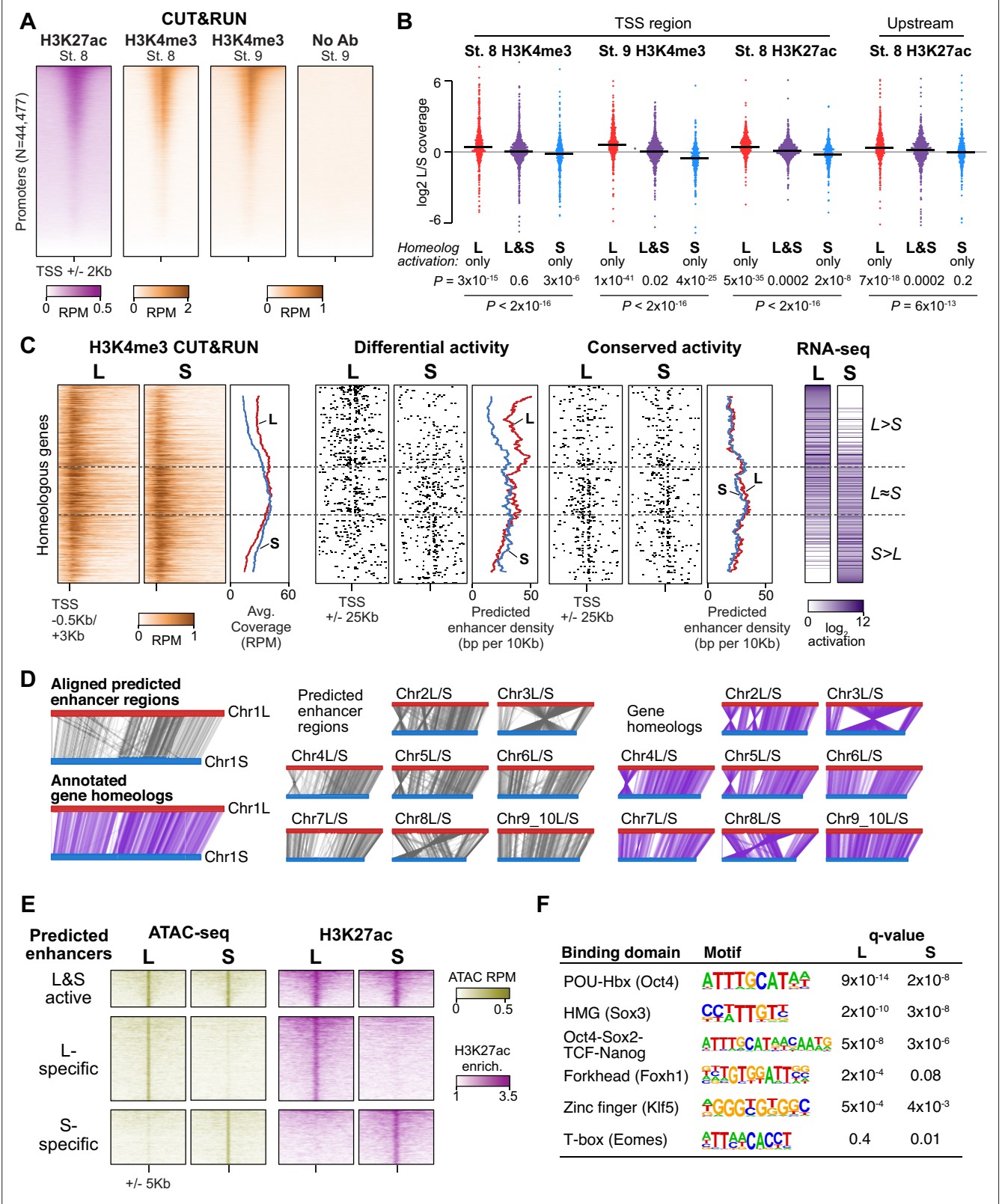

**Figure 3.** Differential homeolog activation is regulated by subgenome-specific enhancers. (**A**) CUT&RUN coverage over all annotated transcription-start site (TSS) regions, sorted by descending stage 8 H3K27ac signal. (**B**) Bee-swarm plots showing the log2 ratio of L versus S homeolog coverage among genes where only one homeolog is activated (L only, S only), or both homeologs are activated. TSS region is 1 kb centered on the TSS; upstream region is 500 bp to 3 kb upstream of the TSS. Horizontal bars show medians. Individual category p values are from two-sided paired t-tests of log2 L

*Figure 3 continued on next page*

Figure 3 continued

homeolog coverage vs log2 S homeolog coverage, p values across the three categories are from a one-way ANOVA on the log2 ratios. (**C**) Stage 9 H3K4me3 CUT&RUN coverage over paired homeologous gene regions around the TSS (left) and maps comparing high-confidence predicted enhancer density near homeologous TSSs (middle). Differential predicted enhancers are active in only one subgenome, conserved predicted enhancers are active in both. Average densities are plotted to the right of each paired map. Gene pairs are sorted according to L versus S subgenome RNA-seq activation ratio (right). (**D**) Schematics showing aligned predicted enhancers and their homeologous regions (gray) mapped onto L (red, top lines) and S (blue, bottom lines) chromosomes. Comparable schematics show Xenbase annotated homeologous gene pairs (lavender). (**E**) Heatmap of stage 9 ATAC-seq and stage 8 H3K27ac CUT&RUN over L & S homeologous regions for equivalently active high-confidence predicted enhancers (top) and subgenome-specific predicted enhancers. (**F**) Top enriched transcription factor motif families in L-specific and S-specific active high-confidence predicted enhancers compared to inactive homeologous regions. FDR-corrected p-values from Homer are shown. RPM = reads per million.

The online version of this article includes the following figure supplement(s) for figure 3:

**Figure supplement 1.** Profiling homeologous regulatory elements.

**Figure supplement 2.** Enhancer prediction and homeologous region comparison.

Instead, we searched for differences in gene-distal regulatory elements – that is enhancers – between the two subgenomes. To identify regions of open chromatin characteristic of enhancers, we performed Assays for Transposase-Accessible Chromatin with sequencing (ATAC-seq) on dissected animal cap explants; the high concentration of yolk in vegetal cells inhibits the Tn5 transposase (*Esmaeili et al., 2020*). Accessible chromatin is already evident at stage 8 in putative enhancer regions, though the overall signal is weak, and by stage 9, these regions exhibit robust accessibility (*Figure 3—figure supplement 2A*). We called peaks of elevated sub-nucleosome sized fragment coverage at stage 9, then intersected the open regions with our H3K27ac CUT&RUN. This yielded 15,654 putative open and acetylated gene-distal regulatory regions at genome activation, of which we classified 5047 as high confidence predicted enhancers that had ≥2 fold signal enrichment in each of at least three H3K27ac replicates and three ATAC-seq replicates individually (*Figure 3—figure supplement 2A*, *Supplementary file 5*).

To identify homeologous L and S enhancer regions, we constructed a subgenome chromosome-chromosome alignment using LASTZ (*Harris, 2007*). This yielded a syntenic structure consistent with genetic maps (*Figure 3D*; *Session et al., 2016*), recapitulating the large inversions between chr3L/chr3S and chr8L/chr8S. Seventy-nine percent of predicted enhancer regions successfully lifted over to homeologous chromosomes, and of these, >92% of these are flanked by the same homeologous genes (*Figure 3—figure supplement 2B*), confirming local synteny.

Among the paired regions involving high-confidence predicted enhancers, only 21% had conserved activity in both homeologs, with the remaining pairs exhibiting differential H3K27ac and chromatin accessibility (*Figure 3E*, *Figure 3—figure supplement 2C*). Differential predicted enhancer density around genes significantly correlated with differential activation (p=1.3 × 10$^{-16}$, Pearson's correlation test; *Figure 3C*, middle, *Figure 3—figure supplement 2D*), with greater L enhancer density around differentially activated L genes, and similarly for S enhancers and S genes. In contrast, conserved enhancers had equivalent density near both homeologs regardless of activation status (p=0.67, Pearson's correlation test; *Figure 3C*, right). Thus, differences in enhancer activity likely underlie divergent gene homeolog transcription at genome activation.

## Maternal pluripotency factors differentially engage the subgenomes

Given that these paired enhancer regions are differentially active despite having similar base sequences, we searched for transcription factor binding motifs that distinguished active enhancers from their inactive homeolog. Two motifs were strongly enriched in both active L enhancers and active S enhancers, corresponding to the binding sequences of the pluripotency factors OCT4 and SOX2/3 (SOXB1 family; *Figure 3F*, *Supplementary file 4*). Since mammalian OCT4 and SOX2 are master regulators of pluripotent stem cell induction (*Li and Belmonte, 2017*; *Takahashi and Yamanaka, 2016*), and zebrafish homologs of these factors are maternally provided and required for embryonic genome activation (*Lee et al., 2013*; *Leichsenring et al., 2013*; *Miao et al., 2022*), we hypothesized that differential enhancer binding by maternal *X. laevis* OCT4 and SOXB1 homologs underlies asymmetric activation of the L and S subgenomes.

RNA-seq confirms high maternal levels of *sox3* and *pou5f3.3* (OCT4 homolog) mRNA, as well as lower levels of paralog *pou5f3.2*, each deriving from both subgenomes (*Figure 3—figure supplement*

*2E*). To assess their roles in genome activation, we inhibited their translation using previously validated antisense morpholinos (*Morrison and Brickman, 2006*; *Takebayashi-Suzuki et al., 2007*; *Zhang et al., 2003*) injected into stage 1 embryos. Combinations of *pou5f3.3+sox3* morpholinos and *pou5f3.2+pou5 f3.3* morpholinos led to mild and severe gastrulation defects, respectively, while combining all three morpholinos led to developmental arrest with a complete failure to close the blastopore (*Figure 4—figure supplement 1A*), consistent with what has been reported in *X. tropicalis* (*Gentsch et al., 2019*).

RNA-seq of morpholino-treated embryos at stage 9 revealed extensive misregulation of genome activation, though only 15% of genes exhibited deficient activation, while 43% of genes actually exhibited slightly higher levels in the morphants (*Figure 4A*, *Figure 4—figure supplement 1B and G–K*), which could be due to direct or indirect transcriptional repression mediated by Pou5f3 and Sox3. Increases were predominantly detected from intron signal (*Figure 4—figure supplement 1J*), which would largely rule out post-transcriptional effects. A larger proportion of strictly zygotic genes were down-regulated in the morphants compared to maternal-to-zygotic genes (p=6.6 × 10-18, $\chi$-squared test, 3 d.o.f.), perhaps reflecting a more complex regulatory network that regulates maternal gene reactivation (*Figure 4—figure supplement 1L*).

To further clarify the regulatory network, we also performed morpholino treatments followed by cycloheximide treatment at stage 8, collecting at stage 9 for RNA-seq, to focus the loss of function on primary activation. In these embryos, nearly 70% of first-wave genes were down regulated, including the *mir-427* transcript (*Figure 4A–C*, *Figure 4—figure supplement 1B–F, I*), suggesting that maternal Pou5f3 and Sox3 directly activate a large proportion of first-wave genome activation, but newly synthesized zygotic factors rapidly mobilize to refine target gene expression levels (*Figure 4H*).

In the absence of wild-type Pou5f3 and Sox3 activity, divergent homeolog activation is reduced for a subset of genes, indicating that these factors at least partially underlie differential subgenome activation (*Figure 4D–G*). Among primary-activated genes, there does not seem to be a strong bias toward greater regulation of either homeolog; however, a significantly larger proportion of strictly zygotic genes encoded on both subgenomes is activated by Pou5f3 and Sox3 compared to singleton genes (p=0.0020, $\chi$-squared test, 5 d.o.f.; *Figure 4—figure supplement 1M, N*), which may reflect a reliance on these factors to mediate homeolog-specific expression when two copies exist.

To interrogate Pou5f3 and Sox3 chromatin binding across the subgenomes, we performed CUT&RUN on stage 8 embryos injected with mRNA encoding V5 epitope-tagged *pou5f3.3.L* and *sox3.S*. Peak calling revealed thousands of binding sites for each factor (*Figure 4—figure supplement 2A–D*), and Homer de novo motif analysis recovered the OCT4 and SOX3 binding sequences as top hits (p=$10^{-252}$ and p=$10^{-104}$, respectively; *Figure 4I*, *Figure 4—figure supplement 2E and F*). A subset of peaks have CUT&RUN enrichment for both factors, and at least 10% of peaks contain matches to the Oct4-Sox2 heterodimer motif (*Figure 4—figure supplement 2F*), suggesting Pou5f3 and Sox3 may form a complex in the blastula, similar to mammalian OCT4 and SOX2 (*Boyer et al., 2005*; *Dailey and Basilico, 2001*). CUT&RUN signal for both factors is enriched in the vicinity of genes down-regulated in morphants with or without cycloheximide treatment, but notably not for genes up-regulated (p<1 × $10^{-43}$, Kruskal-Wallis test; *Figure 4I and J*), confirming that up regulation is likely an indirect effect of Pou5f3/Sox3 loss of function.

Down-regulated genes are highly significantly nearer to predicted regulatory elements with enriched Pou5f3 and Sox3 binding (p<1 × $10^{-300}$, Kruskal-Wallis test; *Figure 4K*, *Figure 4—figure supplement 2G*). Differential Pou5f3 and Sox3 binding mirrors differential predicted enhancer activity (*Figure 4—figure supplement 2I*), and subgenome-specific Pou5f3 and Sox3 binding is enriched in the vicinity of the homeolog affected by Pou5f3/Sox3 loss of function (p=7.9 × $10^{-13}$, Kruskal-Wallis test; *Figure 4L*, *Figure 4—figure supplement 2H*). Together, these results implicate Pou5f3.3 and Sox3 in regulating ZGA differentially between the two subgenomes.

## The ancestral pluripotency program is maintained, despite enhancer turnover

Finally, to understand differential activation given the natural history of *X. laevis* allotetraploidy, we compared *X. laevis* subgenome activation patterns to diploid *X. tropicalis* as a proxy for the ancestral *Xenopus*, since there are no known extant diploid descendants of either *X. laevis* progenitor (*Session et al., 2016*). For three-way homeologs/orthologs that are strictly zygotic in *X. laevis*, there is broad

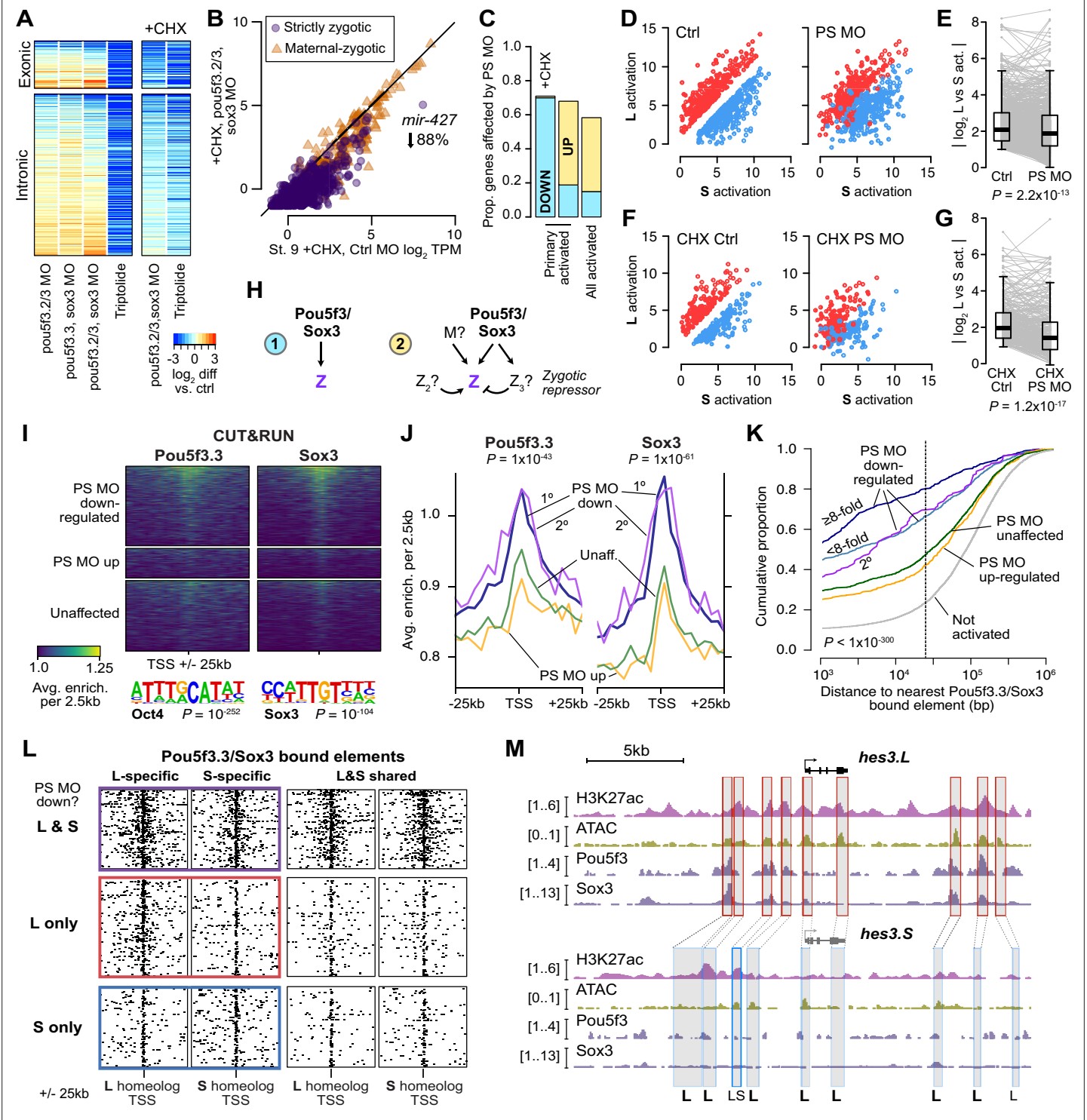

**Figure 4.** Pou5f3.3 and Sox3 binding drives genome activation. (**A**) Heatmap showing log2 fold activation differences for exonic and intronic regions of primary-activated genes for combinations of *pou5f3.2, pou5f3.3,* and *sox3* morpholino-treatments, or Triptolide treatment, compared to controls. Right panel is in the presence of cycloheximide (CHX). (**B**) Biplot comparing exonic expression levels in cycloheximide-treated control embryos versus embryos also injected with *pou5f3.2, pou5f3.3,* and *sox3* morpholinos. Primary-activated genes with maternal contribution <1 TPM (strictly zygotic) are purple circles, maternal-zygotic genes detected by exonic increases are orange triangles. TPM = transcripts per million. (**C**) Barplot summarizing the proportion of genes affected by morpholino treatment with cycloheximide on primary-activated genes (left bar), without cycloheximide (middle bar), and all stage 9 activated genes without cycloheximide (right bar). Down = significantly decreased in one of the morpholino treatments, up = significantly increased. (**D, F**) Biplots showing genes with >2 fold L or S biased activation (upper red and lower blue points, respectively) in control

*Figure 4 continued on next page*

*Figure 4 continued*

embryos (left panel) versus their activation in *pou5f3.2*, *pou5f3.3*, and *sox3* morpholino-treated embryos (right panel, maintaining the same color per gene). (**E, G**) Quantification of the biplots in (**D, F**) in before-and-after plots. Y-axis is the absolute value of the log2 L vs S activation difference. p Values are from Wilcoxon signed-rank tests (paired). Overlaid boxplots show median, upper and lower quartiles, and 1.5 x interquartile range. (**H**) Regulatory networks consistent with direct regulation of embryonic gene activation by Pou5f3 and Sox3 (1) versus additional regulation by zygotic factors (2), which likely accounts for genes up-regulated in MO treatments. (**I**) Stage 8 Pou5f3.3 (left) and Sox3 (right) CUT&RUN coverage near TSSs for genes down-regulated in morpholino-treated embryos with or without cycloheximide (top), genes up-regulated (middle), and genes not significantly affected in any morpholino treatment (bottom). Top enriched motifs for each factor are shown below with p-values from Homer de novo discovery. (**J**) Aggregate plots of the binding signal in (**I**), with down-regulated genes further separated into genes down-regulated with morpholino treatment and cycloheximide (1°) or only down-regulated without cycloheximide (2°). p Values are from Kruskal-Wallis tests on summed signal per TSS. (**K**) Cumulative distributions of distance from a Pou5f3/Sox3-bound regulatory element for genes strongly ($\geq$8 fold) and less strongly (<8 fold) down-regulated in morpholino-treated embryos with or without cycloheximide, compared to up-regulated, unaffected and unactivated genes. p Value is from a Kruskal-Wallis test. (**L**) Maps showing density of Pou5f3/Sox3-bound regulatory elements around paired homeologous TSSs, divided into elements with differential homeologous L & S binding (left panels) versus both bound (right panels). TSSs are grouped according to L versus S homeolog sensitivity to morpholino treatment. (**M**) Browser tracks showing CUT&RUN enrichment and ATAC-seq coverage near active homeolog *hes3.L* and inactive homeolog *hes3.S*. Seven L-specific high-confidence regulatory regions are highlighted with their homeologous S regions (bold 'L'), as well as two lower-confidence enhancers, one of which also has weak activity in S, but minimal Pou5f3 or Sox3 binding (labeled 'LS').

The online version of this article includes the following figure supplement(s) for figure 4:

**Figure supplement 1.** Assessing Pou5f3 and Sox3 roles in genome activation.

**Figure supplement 2.** Pou5f3 and Sox3 CUT&RUN.

conservation of relative expression levels between the *X. tropicalis* and *X. laevis* embryonic transcriptomes after genome activation, when *X. laevis* homeolog levels are summed gene-wise (Spearman's $\rho$=0.67) (***Figure 5A***, left, ***Figure 5—figure supplement 1A–C***, ***Supplementary file 6***). However, the correlation weakens when the *X. laevis* subgenomes are considered independently: relative activation levels in one subgenome alone are depressed relative to *X. tropicalis*, with expression of some genes completely restricted to one subgenome or the other (L, Spearman's $\rho$=0.56; S, $\rho$=0.47; ***Figure 5A***, middle, right, ***Figure 5—figure supplement 1A–C***). Indeed, the *X. tropicalis* embryonic transcriptome is a better estimator for the composite *X. laevis* transcriptome than for either subtranscriptome individually (p<4.3 × 10$^{-13}$ for strictly zygotic genes, p<1.4 × 10$^{-83}$ for maternal-zygotic genes, Wilcoxon signed-rank test on residuals; ***Figure 5—figure supplement 1B***). If the diploid L and S progenitor embryos each exhibited the inferred ancestral activation levels, then these trends strongly suggest that *X. laevis* underwent regulatory changes post allotetraploidization that maintained relative gene expression dosage for embryonic genome activation.

Most activated genes also have a maternal contribution, which can offset asymmetries in homeolog activation levels (***Figure 5—figure supplement 1C***); and indeed, most *X. laevis* genes without conserved activation in *X. tropicalis* nonetheless have conserved embryonic expression due to the maternal contribution (***Figure 5B***). When we specifically compare zygotic activation between the two species, genes activated from both *X. laevis* homeologs are more likely to have orthologous *X. tropicalis* activation (p=1.7 × 10$^{-44}$, $\chi$-squared test, 4 d.o.f.; ***Figure 5B***), as well as conserved *X. tropicalis* Pou5f3/Sox3 regulation (p<1.7 × 10$^{-6}$, $\chi$-squared test, 8 d.o.f.; ***Figure 5—figure supplement 1D***; ***Gentsch et al., 2019***). This suggests that many differentially activated homeologs have acquired novel ZGA regulation by Pou5f3 and Sox3 compared to the inferred ancestral state. Predicted enhancers also follow this trend: subgenome-specific predicted enhancers are less likely to be conserved with *X. tropicalis* (p=1.5 × 10$^{-76}$, $\chi$-squared test for high confidence enhancers, 4 d.o.f; ***Figure 5C***, ***Figure 5—figure supplement 1E***), consistent with a greater degree of regulatory innovation underlying differentially activated homeologs.

This trend is also apparent at greater evolutionary distances. We find that genes activated in *X. laevis* are largely also expressed in zebrafish embryos (~450 million years separated) (***Figure 5—figure supplement 1F***). Despite considerable divergence in activation timing, co-activated *X. laevis* homeologs are still more likely to be part of the first wave of zebrafish genome activation (p=4.3 × 10$^{-12}$, $\chi$-squared test, 4 d.o.f.) and targeted by zebrafish maternal homologs of OCT4 and SOX2, but also NANOG (p=1.8 × 10$^{-45}$, $\chi$-squared test, 6 d.o.f.) (***Figure 5D***, ***Figure 5—figure supplement 1G***). Subgenome-shared predicted enhancers are also more likely to have evidence for conservation in zebrafish (p=2.8 × 10$^{-58}$ for all enhancers, p=5.0 × 10$^{-4}$ for high-confidence enhancers, $\chi$-squared tests, 4 d.o.f.) (***Figure 5—figure supplement 1H***; ***Bogdanovic et al., 2012***). Taken together, this

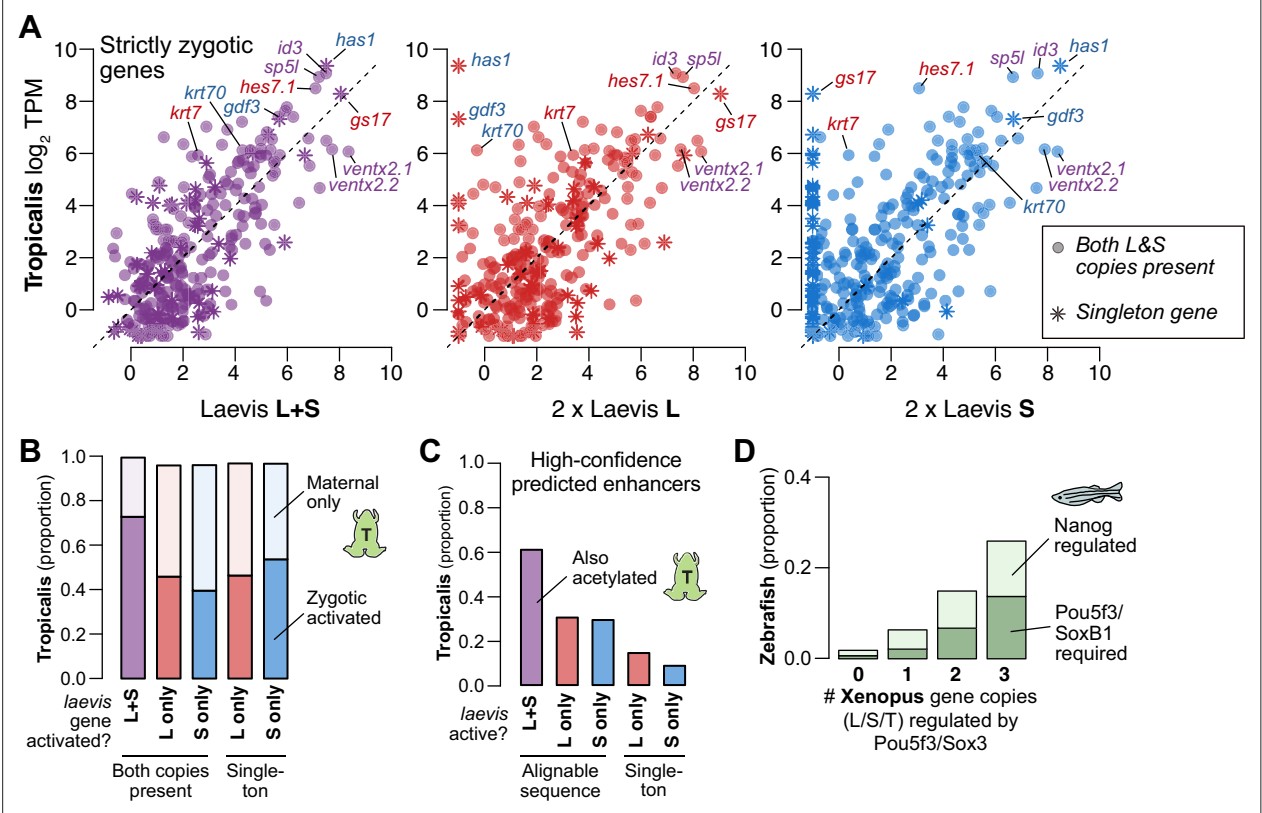

**Figure 5.** Regulatory divergence underlies dosage maintenance. (**A**) Biplots comparing relative expression levels of activated genes in *X. laevis* and *X. tropicalis*, treating L and S homeolog contributions separately (middle, right) or summed (left). Individual subgenome expression is scaled 2 x, since transcript per million (TPM) normalization is calculated relative to the entire *X. laevis* transcriptome. Individual labeled genes are color coded according to the dominant expressed homeolog (red = L, blue = S, purple = equivalent). (**B**) Barplots showing the proportion of *X. laevis* genes across homeolog activation categories whose orthologs are also activated in *X. tropicalis* or part of the maternal contribution. (**C**) Barplots showing the proportion of *X. laevis* enhancers across homeolog activity categories that are acetylated in *X. tropicalis*. (**D**) Barplots showing the proportion of *Xenopus* genes whose orthologs are regulated by Pou5f3/SoxB1 and Nanog in zebrafish. *Xenopus* genes are classified according to how many homeo/orthologs are regulated by Pou5f3/Sox3. Genes with conserved regulation in both *X. laevis* homeologs and *X. tropicalis* are more likely to be regulated by Pou5f3/SoxB1 in zebrafish, but also more likely to be regulated by Nanog.

The online version of this article includes the following figure supplement(s) for figure 5:

**Figure supplement 1.** Shared patterns of activation with other taxa.

suggests that the regulatory architecture underlying differential homeolog activation in *X. laevis* is more likely to be derived, in contrast to the deeply conserved networks that regulate many co-activated homeologs.

Interestingly, *Xenopus* and possibly all Anuran amphibians lack a NANOG ortholog, likely due to a chromosomal deletion (*Schuff et al., 2012*). In the absence of a Nanog homolog in the maternal contribution, we find that maternal Pou5f3.3 and Sox3 seem to have subsumed NANOG's roles in *X. laevis* genome activation, while zygotic factors such as Ventx help promote cell potency in the early gastrula (*Scerbo et al., 2012*; *Schuff et al., 2012*). This demonstrates core-vertebrate mechanistic conservation in genome activation amid both cis- and trans-regulatory shuffling, which converge to support pluripotent stem cell induction and embryonic development.

## Discussion

Together, our findings establish the pluripotency factors Pou5f3.3 and Sox3 as maternal activators of embryonic genome activation, which are differentially recruited to the two homeologous subgenomes of *X. laevis* by a rewired enhancer network (*Figure 6*). Of the thousands of genes activated during the MZT, a majority of annotated homeolog pairs experience differential activation, which appears to be

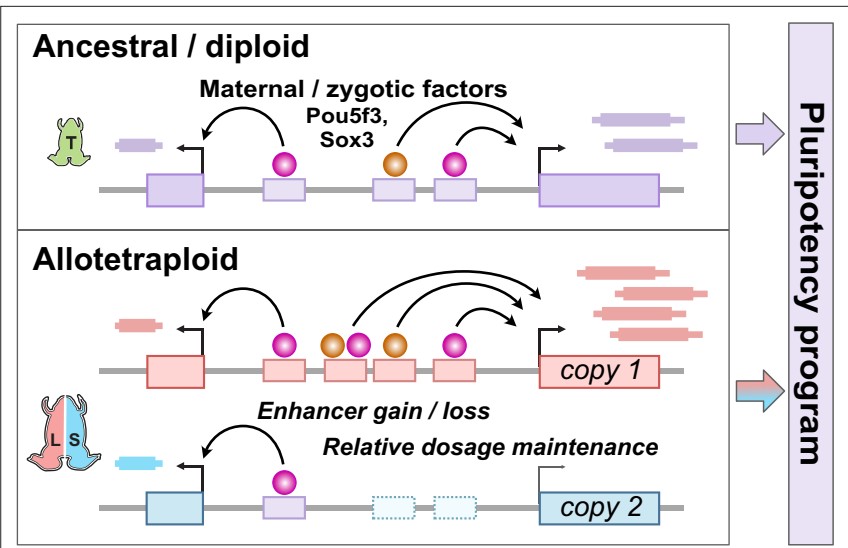

**Figure 6.** Model for pluripotency network evolution. *X. laevis* likely underwent extensive enhancer turnover between its two subgenomes, which nonetheless maintained stoichiometry of pluripotency reprogramming in the early embryo.

driven by subgenome-specific enhancer gain and/or loss correlated with differential Pou5f3.3/Sox3 binding and regulation. However, this magnitude of regulatory divergence seems to have had a net neutral effect, as combined subgenome activation produces a composite reprogrammed embryonic transcriptome akin to diploid *X. tropicalis*.

As embryogenesis proceeds, regulatory divergence between the subgenomes is likely even broader. In *X. tropicalis*, signal transducers and transcription factors including Pou5f3.2/3, Sox3, Smad1/2, β-catenin, Vegt, Otx1, and Foxh1 regulate embryo-wide and regional gene activation (*Charney et al., 2017*; *Gentsch et al., 2019*; *Paraiso et al., 2019*), and binding motifs for some of these are found in differentially active *X. laevis* enhancers (*Figure 3F*, *Supplementary file 4*). Additionally, by focusing on accessible chromatin in animal caps, we may have underestimated the magnitude of homeologous enhancer divergence regulating endodermal fate in the vegetal cells. But based on the close morphological similarity of *X. tropicalis* and *X. laevis* embryos, we would predict that these subgenome regulatory differences also converge to producing ancestral dosages in the transcriptome.

Although homeolog expression bias can derive from gene regulatory differences evolved in the parental species prior to hybridization (*Buggs et al., 2014*; *Grover et al., 2012*), we propose that regulatory upheaval in *X. laevis* post-hybridization (i.e. 'genome shock' *McClintock, 1984*) led to expression level gain or loss in one homeolog, which was subsequently corrected by compensatory changes to the other homeolog, possibly repeatedly (*Shi et al., 2012*; *Tirosh et al., 2009*). This implies that early development exerts constraint on the reprogrammed embryonic transcriptome while tolerating (or facilitating) regulatory turnover. The apparent reconfiguration of the *mir-427* cluster after the *X. laevis* and *tropicalis* lineages split similarly highlights how essential MZT regulatory mechanisms can evolve, ostensibly neutrally given that *miR-427*-directed maternal clearance is conserved in *Xenopus*. Thus, *X. laevis* embryos illustrate how the pluripotency program may have accommodated regulatory network disruptions, genomic instability, and aneuploidy across the animal tree.

## Methods
### Animal husbandry
All animal procedures were conducted under the supervision and approval of the Institutional Animal Care and Use Committee at the University of Pittsburgh under protocol #21120500. *Xenopus laevis* adults (Research Resource Identifier NXR_0.0031; NASCO) were housed in a recirculating aquatic system (Aquaneering) at 18 °C with a 12/12 hr light/dark cycle. Frogs were fed 3 x weekly with Frog Brittle (NASCO #SA05960 (LM)M).

## Embryo collection

Sexually mature females were injected with 1000 IU human chorionic gonadotropin into their dorsal lymph sac and incubated overnight at 16 °C. Females were moved to room temperature to lay. Eggs from two mothers per collection were pooled and artificially inseminated using dissected testes in MR/3 (33 mM NaCl, 0.6 mM KCl, 0.67 mM CaCl$_2$, 0.33 mM MgCl$_2$, 1.67 mM HEPES, pH 7.8; *Sive and Richard, 2000*). Dissected testes were stored up to one week in L-15 medium at 4 °C prior to use. Zygotes were de-jellied (*Sive and Richard, 2000*) in MR/3 pH 8.5, with 0.3% β-mercaptoethanol with gentle manual agitation, neutralized with MR/3 pH 6.5, washed twice with MR/3 and incubated in MR/3 at 23 °C until desired developmental stage based on morphology, for genomics experiments.

## RNA-seq libraries

All stage 9 embryos were collected halfway through the stage, at 8 hours post fertilization ('stage 9.5'). Stage 10.5 embryo libraries were spiked with GFP, mCherry, and luciferase in vitro transcribed RNA for an unrelated purpose. Triptolide samples were bathed in 20 µM triptolide in DMSO (200 X stock added to MR/3) at stage 1 and cycloheximide samples were bathed in 500 µg/mL cycloheximide in DMSO at the beginning of stage 8; both were collected when batch-matched, untreated embryos were halfway through stage 9. Equivalent volumes of DMSO were used to treat control samples. Previously validated morpholinos targeting *pou5f3.2* (AGGGCTGTTGGCTGTACATGGTGTC) (*Takebayashi-Suzuki et al., 2007*) *pou5f3.3* (GTACAATATGGGCTGGTCCATCTCC) (*Morrison and Brickman, 2006*) and *sox3* (AACATGCTATACATTTGGAGCTTCA) (*Zhang et al., 2003*), along with control GFP morpholino (ACAGCTCCTCGCCCTTGCTCACCAT) were ordered from GeneTools. Morpholino treated embryos were injected at stage 1 with *pou5f3.3*, *sox3*, and/or GFP control morpholino. Non-cycloheximide treated embryos all received 120 ng total morpholino, consisting of 40 ng of each target morpholino augmented with 40 ng of GFP morpholino for two-morpholino conditions. Cycloheximide-treated embryos received 40 ng of each target morpholino for the triple condition, 40 ng *pou5f3.3*+40 ng GFP, 40 ng *sox3* +40 ng GFP, 40 ng *pou5f3.3*+40 ng sox3, or 80 ng GFP. An additional cycloheximide-treated high concentration morpholino condition used 55 ng *pou5f3.3*+75 ng *sox3*. Each embryo was injected twice with 5 nl of MO on opposite sides. Embryos were allowed to recover to stage 5 before moving to MR/3 to develop, and collected when batch-matched, untreated embryos were halfway through stage 9. Samples from the 'H' batch likely had an issue with the cycloheximide treatment, based on greater similarity of gene expression to untreated samples than other cycloheximide-treated samples, and were removed from subsequent analyses.

For phenotype observation, embryos were incubated at 23 °C or 18 °C after injection and photographed when control embryos reached stage 10.5 for 23 °C and stage 12 for 18 °C. For RNA extraction, two embryos per sample were snap frozen and homogenized in 500 µl of TRIzol Reagent (Invitrogen #15596026) followed by 100 µl of chloroform. Tubes were spun at 18,000 x *g* at 4 °C for 15 min, the aqueous phase was transferred to a fresh tube with 340 µl of isopropanol and 1 µl of GlycoBlue (Invitrogen #AM9515), then precipitated at –20 °C overnight. Precipitated RNA was washed with cold 75% ethanol and resuspended in 50 µl of nuclease-free water. Concentration was determined by NanoDrop.

For library construction, rRNA depletion was performed as per *Phelps et al., 2021* with *X. laevis* specific oligos reported previously: 1 µl of antisense nuclear rRNA oligos and 1 µl of antisense mitochondrial rRNA oligos (final concentration 0.1 µM per oligo) were combined with 1 µg of total RNA in a 10 µl buffered reaction volume (100 mM Tris-HCl pH 7.4, 200 mM NaCl, 10 mM DTT), heated at 95 °C for 2 minutes and cooled to 22 °C at a rate of 0.1 °C/s in a thermocycler. Next, 10 U of thermostable RNaseH (NEB #M0523S) and 2 µl of provided 10 X RNaseH buffer were added and volume brought to 20 µl with nuclease-free water. The reaction was incubated at 65 °C for 5 or 30 min, then 5 U of TURBO DNase (Invitrogen #AM2238) and 5 µl of provided 10 x buffer was added, volume brought to 50 µl with nuclease-free water and incubated at 37 °C for 30 min. The reaction was purified and size selected to >200 nts using Zymo RNA Clean and Concentrator-5 (Zymo #R1013) according to manufacturer's protocol, eluting in 10 µl of nuclease-free water. The WT Stage 5 sample was also depleted of mitochondrial COX2 and COX3 mRNA as part of the *Phelps et al., 2021* study. Strand-specific RNA-seq libraries were constructed using NEB Ultra II RNA-seq library kit (NEB #E7765) according to manufacturer's protocol with fragmentation in first-strand buffer at 94 °C for 15 min. Following first and second strand synthesis, DNA was purified with 1.8 X AmpureXP beads

(Beckman #A63880), end repaired, then ligated to sequencing adaptors diluted 1:5. Ligated DNA was purified with 0.9 X AmpureXP beads and PCR amplified for 8 cycles, then purified again with 0.9 X AmpureXP beads. Libraries were verified by Qubit dsDNA high sensitivity (Invitrogen #Q32851) and Fragment Analyzer prior to multiplexed sequencing at the Health Sciences Sequencing Core at Children's Hospital of Pittsburgh.

For samples used for differential expression analysis, separate libraries were constructed for each of two replicate sets of embryos from each experimental day, which were considered biological replicates for DESeq2. All libraries from the same experimental day are labeled with the same batch designation (e.g. a, b, c,...).

## CUT&RUN

CUT&RUN procedure was adapted from *Hainer and Fazzio, 2019* optimizations of the method of *Skene and Henikoff, 2017*. For nuclear extraction, embryos were de-vitellinized using 1 mg/mL pronase dissolved in MR/3. Once the vitelline envelope was removed, 12–24 embryos (50K – 100K cells) were carefully transferred into 1 mL of NP2.0 buffer (*Briggs et al., 2018*) in a 1.5 mL tube and gently agitated (pipetting buffer over the surface of the embryos) until cells have dissociated. The buffer was carefully drawn off to the level of the cells and 1 mL of Nuclear Extraction (NE) buffer (20 mM HEPES-KOH, pH 7.9, 10 mM KCl, 500 µM spermidine, 0.1% Triton X-100, 20% glycerol) with gentle pipetting with a clipped P1000, and the lysate was centrifuged at 600x*g* in 4 °C for 3 min. The free nuclei were then bound to 300 µL of activated concanavalin A beads (Polysciences #86057) at RT for 10 min. Nuclei were blocked for 5 min at RT then incubated in 1:100 dilution of primary antibody for 2 hr at 4 °C, washed, incubated in a 1:200 dilution of pAG MNase for 1 hr at 4 °C, and washed again. The bound MNase was activated with 2 mM CaCl$_2$ and allowed to digest for 30 min, then stopped using 2 x STOP buffer (200 mM NaCl, 20 mM EDTA, 4 mM EGTA, 50 µg/mL RNase A, 40 µg/mL glycogen). Nuclei were incubated at 37 °C for 20 min followed by centrifuging for 5 min at 16,000x*g*, drawing off the DNA fragments with the supernatant. The extracted fragments were treated with SDS and proteinase K at 70 °C for 10 min followed by phenol chloroform extraction. Purified DNA was resuspended in 50 µL of water and verified by Qubit dsDNA high sensitivity and Fragment Analyzer. Antibodies used were: H3K4me3, Millipore #05–745 R, RRID:AB_1587134, Lot #3257057 (stage 8 rep 1 & stage 9 reps 1 & 2) and Invitrogen #711958, RRID:AB_2848246, Lot #2253580; H3K27ac, ActiveMotif #39135, RRID:AB_2614979, Lot #06419002; V5, Invitrogen #R960-25, RRID:AB_2556564, Lot #2148086. At least three biological replicate libraries from different embryo collection days were constructed for the key samples (St. 8 H3K27ac, St. 9 H3K4me3).

For transcription factor CUT&RUN, *pou5f3.3.L* and *sox3.S* IVT templates were cloned from cDNA using primers for *pou5f3.3.L* – NM_001088114.1 (F: GGACAGCACGGGAGGCGGGGGATCCGAC CAGCCCATATTGTACAGCCAAAC; R: TATCATGTCTGGATCTACGTCTAGATCAGCCGGTCAGGAC CCC) and *sox3.S* - NM_001090679.1 (F: aaaggatccTATAGCATGTTGGACACCGACATCA; R: aaatctag aTTATATGTGAGTGAGCGGTACCGTG) into N-terminal V5-pBS entry plasmids using HiFi assembly (NEB #E2621) for *pou5f3.3* and BamHI/XbaI for *sox3*. IVT was done using NEB HiScribe T7 ARCA kit (#E2065S) on NotI-linearized plasmid for 2 hr at 37 °C, then treated with 5 U of TURBO DNaseI (Invitrogen #AM2238) for 15 min. mRNA was purified using NEB Monarch RNA Cleanup Columns (#T2030) and stored at –80 °C until use. For injection, immediately after dejellying, stage 1 embryos were placed in 4% Ficoll-400 in MR/3. Each embryo was injected with 2.5 nL of 40 ng/µL of mRNA into each cell at stage 3 (4 cell), for a total of 10 nL per embryo. Three biological replicates from different days for each factor were generated. Factor-specific no-antibody CUT&RUN samples were made using the same injected embryos.

CUT&RUN libraries were constructed using the NEB Ultra II DNA library prep kit (NEB #E7645) according to manufacturer's protocol. DNA was end repaired and then ligated to sequencing adaptors diluted 1:10. Ligated DNA was purified with 0.9 x AmpureXP beads and PCR amplified for 15 cycles, then purified again with 0.9 x AmpureXP beads. Libraries were size selected to 175–650 bp for histone modifications and 150–650 bp for transcription factors on a 1.5% TBE agarose gel and gel purified using the NEB Monarch DNA gel extraction kit (#T1020) before being verified by Qubit dsDNA high sensitivity and Fragment Analyzer prior to multiplexed paired-end sequencing on an Illumina NextSeq 500 at the Health Sciences Sequencing Core at Children's Hospital of Pittsburgh.

## ATAC-seq

ATAC procedure was from *Esmaeili et al., 2020* Embryos were grown in MR/3 until desired NF stage and devitellinized individually with fine watch-maker forceps. Ectodermal explants (animal caps) were dissected using watch-maker forceps in 0.7 x MR. Two caps were transferred to 1 mL of ice-cold PBS and centrifuged at 500x*g* in 4 °C for 5 min twice. After washing with PBS, caps were lysed in 50 µl of RSB buffer (10 mM Tris pH 7.4, 10 mM NaCl, 3 mM MgCl2, 0.1% Igepal CA-630) with a clipped P200 pipet. The lysate was centrifuged again for 10 min and the supernatant was drawn off. The pellet was resuspended in 47.5 µl TD buffer (10 mM Tris pH 7.6, 5 mM MgCl2, 10% dimethylformamide) and 2.5 µl of 3 µM transposome (see below) was added. Nuclei were transposed with gentle shaking for 1 hr at 37 ° C before adding 2.5 µl proteinase K and incubating overnight at 37 °C. Transposed DNA was purified using EconoSpin Micro columns (Epoch) and amplified using 25 µM indexed Nextera primers with Thermo Phusion Flash master mix for 12 cycles. Primers used were: CAAGCAGAAGAC GGCATACGAGAT[i7]GTCTCGTGGGCTCGG with i7 indices 707 – gtagagag; 714 –tcatgagc; 716 – tagcgagt; and AATGATACGGCGACCACCGAGATCTACAC[i5]TCGTCGGCAGCGTC with i5 indices 505 – gtaaggag; 510 – cgtctaat; 517 – gcgtaaga; 520 – aaggctat. The amplified library was column cleaned and verified by Qubit dsDNA high sensitivity and Fragment Analyzer and sequenced multi-plexed paired end at the Health Sciences Sequencing Core at Children's Hospital of Pittsburgh. For the first two replicates per stage, after initial sequencing, libraries were subsequently size selected on an agarose gel to enrich for 150–250 and 250–600 bp fragments and resequenced pooled. For a third stage 8 replicate and third and fourth stage 9 replicate, only the 150–250 bp fragments were sequenced. Biological replicate libraries are from different embryo collection days.

Transposomes were constructed according to *Picelli et al., 2014* Adapter duplexes for Tn5ME-A (TCGTCGGCAGCGTCAGATGTGTATAAGAGACAG) +Tn5MErev ([phos]CTGTCTCTTATACACATCT) and Tn5ME-B (GTCTCGTGGGCTCGGAGATGTGTATAAGAGACAG) +Tn5MErev were each annealed in 2 µl of 10 X annealing buffer (100 mM HEPES pH 7.2, 500 mM NaCl, 10 mM EDTA) using 9 µl of each oligo at 100 µM, heated to 95 °C for 1 min then ramped down to 25 °C at 0.1 °C/s in a thermocycler. The two duplexes were held at 25 °C for 5 min then mixed together. On ice, 35 µl of hot glycerol was cooled to 4 °C then 35 µl of the primer mixture and 25 µl of Tn5 (Addgene #112112) was added and mixed and held at 1 hr at RT with gentle pipet mixing every 15 min. Transposomes were stored at –20 °C.

## Transcriptomic analysis

RNA-seq reads were mapped to the *X. laevis* v9.2 genome using HISAT2 v2.0.5 (*Kim et al., 2015*) (--no-mixed `--no-discordant`). Mapped reads were assigned to gene exons (Xenbase v9.2 models) using featureCounts v2.0.1 *Liao et al., 2014* in reversely-stranded paired-end mode with default parameters, and to introns with `--minOverlap` 10 on a custom intron annotation: starting with all introns from the v9.2 GFF file, subtract (a) all regions detected in stage 5 RNA-seq at >2 read coverage, strand specifically; (b) all regions that overlap an annotated exon from a different transcript form; (c) regions that overlap repetitive elements as defined by RepeatMasker (UCSC) and Xenbase-annotated transposons, not strand specifically; (d) regions that ambiguously map to more than one distinct gene's intron (i.e. transcript forms of the same gene are allowed to share an intron, but not between different genes).

DESeq2 v4.0.3 (*Love et al., 2014*) was used for statistical differential expression analysis. To build the DESeq2 model, exon and intron raw read counts were treated as separate rows per gene in the same counts matrix (intron gene IDs were preceded with a 'i_' prefix). Only genes annotated by Xenbase as 'protein_coding', 'lncRNA', or 'pseudogene' were retained. Low-expressed genes were removed (exon reads per million (RPM) <0.5 across all samples) and then low-depth intron features were removed (intron raw read count ≤10 or reads per kilobase per million (RPKM) <0.25 across all samples). Comparisons were made between batch-matched samples where possible, to account for variations in the maternal contribution between mothers. Significant differences with adjusted <0.05 and log2 difference ≥1.5 were used for downstream analysis. High-confidence activated genes had significant increases in DMSO vs Triptolide for both batches and stage 9 vs stage 5. High-confidence primary-activation 'first-wave' genes were high-confidence activated and had significant increase in DMSO vs Cycloheximide. High-confidence activated genes significantly changed in any Pou/Sox morpholino treatment were considered to be affected genes. For chromatin profiling, genes

were classified as Pou/Sox down-regulated if they were significantly decreased in morpholino treatment either with or without cycloheximide. Homeologous genes were paired according to Xenbase GENEPAGE annotations. Genes were considered maternal if they had average stage 5 TPM ≥1. To calculate magnitude of effect for graphing and sorting, the maximal |log2 fold difference| of average exon TPM and average intron RPKM was chosen per gene.

For mir-427 gene identification and RNA-seq coverage visualization, miRBase (*Kozomara et al., 2019*) hairpin sequences MI0001449 and MI0038331 were aligned to the v9.2 and v10.1 reference genomes using UCSC BLAT (*Kent, 2002*) and maximal possible read coverage was graphed allowing all multimappers. To align the v10.1 Chr1L and Chr1S regions flanking the Chr1L mir-427 locus, genomic sequence was extracted between homeologous genes upstream and downstream mir-427. Local alignments with E-value <1e-10 were retained from an NCBI BLAST 2.11.0+blastn alignment (*Camacho et al., 2009*).

dN/dS ratios were calculating using PAML v4.9f (*Yang, 1997*) with L-S pairwise CDS alignments produced by pal2nal v14 (*Suyama et al., 2006*) on amino-acid alignments by EMBOSS needle v6.6.0.0 (-gapopen 10 -gapextend 0.5) (*Rice et al., 2000*).

All other statistical tests were performed using R v4.0.4 (*R Development Core Team, 2013*).

## Chromatin profiling analysis

CUT&RUN and ATAC-seq paired-end reads were mapped to the *X. laevis* v10.1 genome using bowtie2 v2.4.2 (*Langmead and Salzberg, 2012*) (--no-mixed `--no-discordant`) and only high-quality alignments (MAPQ ≥30) were retained for subsequent analysis. Read pairs were joined into contiguous fragments for coverage analyses. For transcription factor CUT&RUN, reads were trimmed using trim_galore v0.6.6 and Cutadapt v1.15 (*Martin, 2011*) in paired-end mode (--illumina `--trim-n`). Downstream analyses were performed using custom scripts with the aid of BEDtools v2.30.0 (*Quinlan and Hall, 2010*), Samtools v1.12 (*Li et al., 2009*), and deepTools v3.5.1 (*Ramírez et al., 2014*).

For promoter-centered analyses, one transcript isoform per gene was selected from Xenbase v9.2 annotations: the most upstream TSS with non-zero RNA-seq coverage at stage 9 was used, otherwise the most upstream TSS if no RNA-seq evidence. Then the corresponding v10.1 coordinates were obtained based on gene name match.

To identify open chromatin regions, aligned stage 9 ATAC-seq fragments pooled between replicates were filtered to <130 bp, then peaks called using MACS2 v2.2.7.1 (*Zhang et al., 2008*) with an effective genome size of 2.74e9 (number of non-N bases in the v10 reference sequence). CUT&RUN no-antibody samples were used as the control sample. To further exclude probable false-positive regions, peaks overlapping any of the following repetitive regions were removed: (a) scRNA, snRNA, snoRNA, or tRNA as annotated by Xenbase; (b) rRNA as determined by BLASTed 45 S, 16 S, 12 S, and 5 S sequences. Peaks on unassembled scaffolds were also excluded.

Putative enhancers had twofold enriched stage 8 H3K27ac CUT&RUN coverage over no antibody, with ≥1 RPKM pooled H3K27ac coverage and <1 RPKM no-antibody coverage, in a 500 bp window centered on ATAC-seq peak summits. A subset of these were additionally annotated as high-confidence ('hi') if they had twofold enrichment in each of at least three individual H3K27ac CUT&RUN samples and three ATAC-seq samples, and lower confidence otherwise ('lo'). Stage 9 ATAC-seq replicates 3 and 4 were pooled to serve as a single sample for this purpose, due to lower read depth. Enhancers were classified as distal ('dist') if they were >1 kb from any Xenbase v10.1 annotated TSS, proximal ('prox') otherwise.

For transcription factor peak calling, replicates were pooled per factor, then individual replicates were verified for enrichment at peaks. No-antibody samples were pooled as a uniform background. MACS2 was run as above, and SEACR v1.3 (*Meers et al., 2019*) was run in norm stringent mode. Peak calls were not used for enhancer analyses; rather, enhancers or homeologous regions with ≥0.5 RPM CUT&RUN coverage and ≥2-fold enrichment over no antibody in a 200 bp window were considered bound.

Coverage heatmaps were generated using deepTools on reads-per-million normalized bigWigs; or enrichment over no-antibody bigWigs generated using deepTools bigwigCompare (--operation ratio `--pseudocount` 0.1 `--binSize` 50 `--skipZeroOverZero`).

For density heatmaps, L/S enhancer pairs were annotated as differential or shared based on one or both partners, respectively, mapping to a putative enhancer, as described above. Pairs were similarly

annotated as differentially or both TF bound based on ≥2-fold enrichment over no antibody for either TF at one or both partners, respectively, and converted to bigWigs representing the genomic location of each bound putative enhancer. Density heatmaps were generated as above and plotted with respect to selected TSSs.

## Motif finding

Enriched sequence motifs in enhancers were identified using Homer v4.11.1 (*Heinz et al., 2010*) in scanning mode against the vertebrate database, using 200 bp of sequence centered on the ATAC-seq peak for enhancers; and 500 bp of sequence centered on the TSS for promoters. Enrichment was calculated using one set of homeologous regions (L or S) as the foreground and the other as the background. The top representative motif per DNA binding domain was reported. For transcription factor peaks, Homer was first used in de novo mode on the top 1000 MACS peaks for Pou5f3 and Sox3 separately; the top motif matched mammalian Oct4 and Sox3 database motifs, respectively. To determine the sequence logo for the Pou5f3-Sox3 dimer motif, a subset of Sox3 peaks with adjacent Pou5f3 and Sox3 motifs was extracted and Homer motif finding was performed using -len 15. To calculate motif prevalence across all peaks, Homer database motif matrices for Oct4 (GSE11431), Sox3 (GSE33059) and OCT4-SOX2-TCF-NANOG (GSE11431) (representing the OCT4-SOX2 dimer motif) were scanned against the entire set of peaks. A set of ATAC-seq accessible regions with no H3K27ac enrichment (rejected regions from the above enhancer prediction analysis) and <0.5 fold enrichment of Pou5f or Sox3 CUT&RUN signal was also scanned to estimate background motif frequencies. Peaks with hits for the dimer motif were secondarily filtered to additionally require the presence of the Pou5f3 and Sox3 individual motifs.

## Homeologous enhancer identification

Each v10.1 chromosome pair (e.g. Chr1L and Chr1S) was aligned using lastZ-1.04.00 (*Harris, 2007*) and UCSC Genome Browser utilities (*Kent et al., 2002*) with parameters adapted from the UCSC Genome Browser previously used to align *X. tropicalis* with *X. laevis* (http://www.bx.psu.edu/miller_lab/dist/README.lastz-1.02.00/README.lastz-1.02.00a.html; http://genomewiki.ucsc.edu/index.php/XenTro9_11-way_conservation_lastz_parameters; no automatic chaining; open = 400, extend = 30, masking = 0, seed = 1 {12of19}, hspthreshold = 3000, chain = 0, ydropoff = 9400, gapped-threshold = 3000, inner = 2000). Chaining and netting were done with axtChain linearGap set to medium and chainSplit lump = 50. Nets were generated using default chainNet and the highest scoring chains were selected from those nets using default netChainSubset. Reciprocal best chains were identified according to UCSC Genome Browser guidelines. The highest scoring chains were reverse referenced, sorted, and then converted to nets using default chainPreNet and chainNet (-minSpace=1 -minScore=0). Reciprocal best nets were selected with default netSyntenic. The new highest scoring best chains were extracted using netChainSubset, converted back to the original reference, and netted as described prior, resulting in reciprocal best, highest scoring chains for use with liftOver.

In the first pass, 500 bp enhancer regions centered on the ATAC-seq peak were lifted to the homeologous subgenome with a 10% minimum sequence match requirement. For enhancers that failed this liftOver, 5 kb enhancer regions were lifted over; as a stringency check, each 2.5 kb half was also individually lifted over, and only regions correctly flanked by both halves were retained. If an enhancer's homeologous region also overlaps an annotated enhancer, it was considered conserved, otherwise it was considered subgenome-specific. To test synteny, the 5 closest Xenbase-annotated genes up- and downstream of each region in a homeologous pair were compared.

## Comparison with *X. tropicalis* and zebrafish

*X. tropicalis* wild-type RNA-seq reads from *Owens et al., 2016*, RiboZero stage 5 (SRA: SRR1795666) and stage 9 (SRA: SRR1795634), were aligned by HISAT2 as above and mapped to Xenbase v10 gene annotations using featureCounts. Pou5f3/Sox3 morpholino and alpha-amanitin-affected genes were obtained from published data tables from *Gentsch et al., 2019*, and the JGI gene accession numbers were mapped to Xenbase GenePage IDs (v7.1). Significantly affected genes were 1.5-fold decreased and adjusted p<0.05. Genes with TPM >1 at either stage 5 or stage 9 were considered embryonic expressed.

For transcriptome comparisons between *X. laevis* subtranscriptomes and *X. tropicalis*, log2 TPM values for non-zero expressed genes per transcriptome (L homologs only, S homologs only, L+S homologs summed, *tropicalis*) were Z-normalized to calculate correlations. To measure gene-wise deviation of the *laevis* transcriptome/sub-transcriptome from *tropicalis*, residuals were calculated using *tropicalis* Z-normalized expression as the predictor variable (i.e. *tropicalis* expression minus *laevis* expression per gene), with the null hypothesis that *tropicalis* is equally as good a predictor for the L+S composite transcriptome compared to L only or S only.

Zebrafish annotations for activated and Pou5f3/Nanog / SoxB1 affected genes were obtained from **Lee et al., 2013** and associated to *Xenopus* genes using Ensembl ortholog annotations (Xenbase to Zfin). First-wave activated zebrafish genes are significantly increased in the U1/U2 spliceosomal RNA inhibited sample over alpha-amanitin (DESeq2 adjusted p<0.05), activated genes are significantly increased by 6 h.p.f. over alpha-amanitin. Pou5f3/SoxB1 affected genes were significantly decreased in the Pou5f3-SoxB1 double loss of function versus wild-type. Nanog-affected genes were significantly decreased in triple loss of function (NSP) but not Pou5f3-SoxB1 double loss of function. Genes with TPM >1 at 2, 4, or 6 h.p.f. were considered embryonic expressed.

To identify putative conserved enhancers in *X. tropicalis*, *X. laevis* enhancers on the v10.1 genome were BLATed (**Kent, 2002**) to the *X. laevis* v9.2 genome, then lifted over to the *X. tropicalis* v9.2 genome using liftOver chains from the UCSC Genome Browser (xenLae2ToXenTro9, 10% minimum sequence match). Successfully lifted over regions were intersected with published *X. tropicalis* H3K27ac stage 9 peaks from **Gupta et al., 2014** that were lifted from the *X. tropicalis* v2 genome to the v9 genome, passing through v7 and requiring 90% minimum sequence match, using liftOver chains from UCSC Genome Browser (xenTro2ToXenTro7 and xenTro7ToXenTro9). *X. laevis* enhancers were lifted over to the zebrafish GRCz11 genome using liftOver chains from the UCSC Genome Browser, passing through *X. tropicalis* (xenLae2ToXenTro9, 10% minimum sequence match; then xenTro9ToXenTro7, 90% minimum sequence match, then xenTro7ToDanRer10, 10% minimum sequence match, then danRer10ToDanRer11 requiring 90% minimum sequence match). Acetylation at zebrafish dome stage was then assessed by intersecting with H3K27ac ChIP-seq peaks from **Bogdanovic et al., 2012** (GEO: GSM915197): reads were aligned to the GRCz11 genome using bowtie2 as above, and peaks called using macs2 as above with an effective genome size of 4.59e8 and no control sample.

## Acknowledgements

We thank S Hainer and lab for providing the pAG-MNase enzyme, assistance with the CUT&RUN protocol, and feedback. We thank M Rebeiz, T Levin, M Turcotte, K Arndt and lab, C Kaplan and lab, and the entire Lee lab for feedback. This project used the University of Pittsburgh Health Sciences Core at UPMC Children's Hospital Pittsburgh for sequencing. This work was supported by the March of Dimes #5-FY16-307, the National Institutes of Health R35GM137973, the Samuel and Emma Winters Foundation, and start-up funds from the University of Pittsburgh to MTL. This research was supported in part by the University of Pittsburgh Center for Research Computing, RRID:SCR_022735, through the resources provided. Specifically, this work used the H2P cluster, which is supported by NSF award number OAC-2117681.

## Additional information

### Funding

| Funder | Grant reference number | Author |
| --- | --- | --- |
| National Institutes of Health | R35GM137973 | Miler T Lee |
| March of Dimes Foundation | 5-FY16-307 | Miler T Lee |
| Samuel and Emma Winters Foundation | | Miler T Lee |

| Funder | Grant reference number | Author |
|---|---|---|

The funders had no role in study design, data collection and interpretation, or the decision to submit the work for publication.

## Author contributions

Wesley A Phelps, Conceptualization, Data curation, Software, Formal analysis, Validation, Investigation, Visualization, Methodology, Writing – original draft, Writing – review and editing; Matthew D Hurton, Investigation, Methodology; Taylor N Ayers, Investigation; Anne E Carlson, Joel C Rosenbaum, Resources; Miler T Lee, Conceptualization, Data curation, Software, Formal analysis, Supervision, Funding acquisition, Validation, Investigation, Visualization, Methodology, Writing – original draft, Project administration, Writing – review and editing

## Author ORCIDs

Wesley A Phelps ⓘ https://orcid.org/0000-0002-4056-2345
Taylor N Ayers ⓘ http://orcid.org/0000-0003-0680-0773
Anne E Carlson ⓘ http://orcid.org/0000-0003-2724-1325
Miler T Lee ⓘ https://orcid.org/0000-0003-0933-0551

## Ethics

All animal procedures were conducted under the supervision and approval of the Institutional Animal Care and Use Committee at the University of Pittsburgh under protocol #21120500.

## Decision letter and Author response

Decision letter https://doi.org/10.7554/eLife.83952.sa1
Author response https://doi.org/10.7554/eLife.83952.sa2

# Additional files

## Supplementary files

- Supplementary file 1. RNA-seq expression values and DESeq2 comparisons.
- Supplementary file 2. Annotations and expression values for activated genes.
- Supplementary file 3. Transcription start site coordinates used for all genes.
- Supplementary file 4. Motif search results.
- Supplementary file 5. Enhancer annotations.
- Supplementary file 6. Comparative genomics with *X. tropicalis* and zebrafish.
- MDAR checklist

## Data availability

All data and analysis files are available with no restrictions on access. Sequencing data are available in the Gene Expression Omnibus (GEO) under accession number GSE207027. Code and auxiliary data files are available on Github, https://github.com/MTLeeLab/xl-zga (copy archived at *Phelps and Lee, 2023*). Additional data files including chromosome alignments are available at OSF, https://osf.io/ct6g8/.

The following datasets were generated:

| Author(s) | Year | Dataset title | Dataset URL | Database and Identifier |
|---|---|---|---|---|
| Phelps WA, Lee MT | 2022 | Hybridization led to a rewired pluripotency network in the allotetraploid *Xenopus laevis* | https://www.ncbi.nlm.nih.gov/geo/query/acc.cgi?acc=GSE207027 | NCBI Gene Expression Omnibus, GSE207027 |
| Phelps WA, Lee MT | 2022 | *Xenopus* MZT | https://doi.org/10.17605/OSF.IO/CT6G8 | Open Science Framework, 10.17605/OSF.IO/CT6G8 |

The following previously published datasets were used:

| Author(s) | Year | Dataset title | Dataset URL | Database and Identifier |
|---|---|---|---|---|
| Bogdanović O, Fernandez-Miñan A, Tena JJ, de la Calle-Mustienes E, Hidalgo C, van Heeringen SJ, Veenstra GJ, Gómez-Skarmeta JL | 2011 | Dynamics of enhancer chromatin signatures mark the transition from pluripotency to cell specification during embryogenesis | https://www.ncbi.nlm.nih.gov/geo/query/acc.cgi?acc=GSE32483 | NCBI Gene Expression Omnibus, GSE32483 |
| Lee MT, Phelps WA | 2020 | Optimized design of antisense oligomers for targeted rRNA depletion | https://www.ncbi.nlm.nih.gov/geo/query/acc.cgi?acc=GSE152902 | NCBI Gene Expression Omnibus, GSE152902 |
| Owens ND, Blitz IL, Lane MA, Patrushev I, Overton JD, Gilchrist MJ, Cho KW, Khokha MK | 2016 | Measuring Absolute RNA Copy Numbers at High Temporal Resolution Reveals Transcriptome Kinetics in Development | https://www.ncbi.nlm.nih.gov/geo/query/acc.cgi?acc=GSE65785 | NCBI Gene Expression Omnibus, GSE65785 |
| Gentsch GE, Smith JC | 2019 | Maternal pluripotency factors initiate extensive chromatin remodelling to predefine first response to inductive signals | https://www.ncbi.nlm.nih.gov/geo/query/acc.cgi?acc=GSE113186 | NCBI Gene Expression Omnibus, GSE113186 |
| Lee MT, Bonneau AR, Giraldez AJ | 2013 | Nanog, SoxB1 and Pou5f1/Oct4 regulate widespread zygotic gene activation during the maternal-to-zygotic transition | https://www.ncbi.nlm.nih.gov/geo/query/acc.cgi?acc=GSE47558 | NCBI Gene Expression Omnibus, GSE47558 |
| Gupta R, Baker JC | 2014 | Enhancer chromatin signatures predict Smad2/3 binding in Xenopus | https://www.ncbi.nlm.nih.gov/geo/query/acc.cgi?acc=GSE56000 | NCBI Gene Expression Omnibus, GSE56000 |
| Johnson K, LaBonne C | 2022 | Transcriptome sequencing of Xenopus laevis animal caps at six time points in transit from pluripotency to 4 lineages: epidermal, neural, ventral mesoderm and endoderm | https://www.ncbi.nlm.nih.gov/geo/query/acc.cgi?acc=GSE198598 | NCBI Gene Expression Omnibus, GSE198598 |

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

# Appendix 1

## Appendix 1—key resources table

| Reagent type (species) or resource | Designation | Source or reference | Identifiers | Additional information |
|---|---|---|---|---|
| Strain, strain background (*Xenopus laevis*) | Nasco wildtype | eNasco | Research Resource #NXR_0.0031 | |
| Antibody | Anti-H3K4me3 recombinant (rabbit polyclonal) | Invitrogen | Cat #711958 RRID #AB_2848246 | CUT&RUN (1:100) |
| Antibody | Anti-H3K4me3 recombinant (rabbit monoclonal) | Millipore | Cat #05–745 R RRID # AB_1587134 | CUT&RUN (1:100) |
| Antibody | Anti-H3K27ac recombinant (rabbit polyclonal) | Active Motif | Cat #39135 RRID #AB_2614979 | CUT&RUN (1:100) |
| Antibody | Anti-V5 (mouse monoclonal) | Invitrogen | Cat #R960-25 RRID #2556564 | CUT&RUN (1:100) |
| Chemical compound, drug | Isethionic acid | Sigma Aldrich | Cat #220078 | |
| Chemical compound, drug | Triptolide | Apexbio | Cat #50-101-1030 | |
| Chemical compound, drug | Cycloheximide | Sigma Aldrich | Cat #01810 | |
| Gene (*Xenopus laevis*) | *pou5f3.3.L* | RefSeq | NM_001088114.1 | Homeolog used for TF CUT&RUN |
| Gene (*Xenopus laevis*) | *sox3.S* | RefSeq | NM_001090679.1 | Homeolog used for TF CUT&RUN |
| Sequence-based reagent | *X. laevis* rRNA depletion oligomers | **Phelps et al., 2021** | DOI: 10.1093/nar/gkaa1072 | |
| Recombinant DNA reagent | Tn5 (plasmid) | Addgene | Cat #112112 | |
| Recombinant DNA reagent | pA/G-MNase (plasmid) | Addgene | Cat #123461 | Purified enzyme gift from S. Hainer |
| Sequence-based reagent | Tn5ME-A | **Picelli et al., 2014** | DOI: 10.1101/gr.177881.114 | |
| Sequence-based reagent | Tn5ME-B | **Picelli et al., 2014** | DOI: 10.1101/gr.177881.114 | |
| Sequence-based reagent | Tn5MErev | **Picelli et al., 2014** | DOI: 10.1101/gr.177881.114 | |
| Sequence-based reagent | Pou5f3.3 morpholino | GeneTools/**Morrison and Brickman, 2006** | DOI: 10.1242/dev.02362 | Targets both homeologs; GTACAATATGGGCTGGTCCATCTCC |
| Sequence-based reagent | Sox3 morpholino | GeneTools/**Zhang et al., 2003** | DOI: 10.1242/dev.00798 | Targets both homeologs; AACATGCTATACATTTGGAGCTTCA |
| Sequence-based reagent | Pou5f3.2 morpholino | Genetools/**Takebayashi-Suzuki et al., 2007** | DOI: 10.1016 /j.mod.2007.09.005 | Targets both homeologs; AGGGCTGTTGGCTGTACATGGTGTC |
| Sequence-based reagent | GFP control morpholino | Genetools | | ACAGCTCCTCGCCCTTGCTCACCAT |
| Sequence-based reagent | Hi-Fi F primer for Pou5f3.3.L ORF | This paper | | GGACAGCACGGGAGGCGGGGGATCC GACCAGCCCATATTGTACAGCCAAAC |
| Sequence-based reagent | Hi-Fi R primer for Pou5f3.3.L ORF | This paper | | TATCATGTCTGGATCTACGTCTAGAT CAGCCGGTCAGGACCCC |
| Sequence-based reagent | F primer for Sox3.S ORF | This paper | | aaaggatcc TATAGCATGTTGGACACCGACATCA |
| Sequence-based reagent | R primer for Sox3.S ORF | This paper | | aaatctaga TTATATGTGAGTGAGCGGTACCGTG |
| Commercial assay, kit | Ultra II RNA library build kit | NEB | Cat #E7760 | |
| Commercial assay, kit | Ultra II DNA library build kit | NEB | Cat #E7645 | |
| Commercial assay, kit | RNA Clean and Concentrator-5 | Zymo | Cat #R1013 | |
| Software, algorithm | Bowtie2 | **Langmead and Salzberg, 2012**; http://bowtie-bio.sourceforge.net/bowtie2 | DOI: 10.1038/nmeth.1923 | v2.4.2 |

*Appendix 1 Continued on next page*

*Appendix 1 Continued*

| Reagent type (species) or resource | Designation | Source or reference | Identifiers | Additional information |
|---|---|---|---|---|
| Software, algorithm | Hisat2 | *Kim et al., 2015*; http://daehwankimlab.github.io/hisat2/ | DOI: 10.1038/nmeth.3317 | v2.0.5 |
| Software, algorithm | featureCounts | *Liao et al., 2014*; https://subread.sourceforge.net/ | DOI: 10.1093/bioinformatics/btt656 | v2.0.1 |
| Software, algorithm | SEACR | *Meers et al., 2019*; https://github.com/FredHutch/SEACR | DOI: 10.1186 /s13072-019-0287-4 | v1.3 |
| Software, algorithm | MACS2 | *Zhang et al., 2008*; https://github.com/taoliu/MACS | DOI: 10.1186 /gb-2008-9-9-r137 | v2.2.7.1 |
| Software, algorithm | BEDtools | *Quinlan and Hall, 2010*; https://bedtools.readthedocs.io/en/latest/ | DOI: 10.1093/bioinformatics/btq033 | v2.30.0 |
| Software, algorithm | DESeq2 | *Love et al., 2014*; https://bioconductor.org/packages/release/bioc/html/DESeq2.html | DOI: 10.1186 /s13059-014-0550-8 | v4.0.3 |
| Software, algorithm | LiftOver | *Kent et al., 2002*; https://hgdownload.soe.ucsc.edu/downloads.html#utilities_downloads | DOI: 10.1101/gr.229102 | |
| Software, algorithm | BLAT | *Kent et al., 2002*; https://hgdownload.soe.ucsc.edu/downloads.html#utilities_downloads | DOI: 10.1101/gr.229102 | |
| Software, algorithm | Blast | *Camacho et al., 2009*; https://blast.ncbi.nlm.nih.gov/doc/blast-help/downloadblastdata.html | DOI: 10.1186/1471-2105-10-421 | v2.11.0+ |
| Software, algorithm | Samtools | *Li et al., 2009*; http://www.htslib.org | DOI: 10.1093/bioinformatics/btp352 | v1.12 |
| Software, algorithm | deeptools | *Ramírez et al., 2014*; https://github.com/deeptools/deepTools | DOI: 10.1519/JSC.0b013e3182a1f44c | v3.5.1 |
| Software, algorithm | LastZ | *Harris, 2007*; https://github.com/lastz/lastz | | v1.04.00 |
| Software, algorithm | Homer | *Heinz et al., 2010*; http://homer.ucsd.edu/homer/download.html | DOI: 10.1016 /j.molcel.2010.05.004 | v4.11.1 |
| Software, algorithm | Paml | *Yang, 1997*; https://github.com/abacus-gene/paml | DOI: 10.1093/bioinformatics/13.5.555 | v4.9f |
| Software, algorithm | pal2nal | *Suyama et al., 2006*; http://www.bork.embl.de/pal2nal/ | DOI: 10.1093/nar/gkl315 | v14 |
| Software, algorithm | EMBOSS needle | *Rice et al., 2000*; https://emboss.sourceforge.net/download/ | DOI: 10.1016 /s0168-9525(00)02024–2 | v6.6.0 |
| Software, algorithm | R | R core team, 2013; https://www.r-project.org | | v4.0.4 |
| Software, algorithm | Trim_galore | *Martin, 2011*; https://github.com/FelixKrueger/TrimGalore | DOI: 10.14806/ej.17.1.200 | v0.6.6 |
| Software, algorithm | Cutadapt | *Martin, 2011*; https://github.com/marcelm/cutadapt | DOI: 10.14806/ej.17.1.200 | v1.15 |
| Other | *X. laevis* genome | Xenbase; https://www.xenbase.org/xenbase/static-xenbase/ftpDatafiles.jsp | *X. laevis* v9.2 genome assembly | Genomic resource. v9.2 |
| Other | *X. laevis* genome | Xenbase; https://www.xenbase.org/xenbase/static-xenbase/ftpDatafiles.jsp | *X. laevis* v10.1 genome assembly | Genomic resource. v10.1 |
| Other | *X. laevis* gene models | Xenbase; https://www.xenbase.org/xenbase/static-xenbase/ftpDatafiles.jsp | *X. laevis* v9.2 gene models | Genomic resource. v9.2 |
| Other | *X. laevis* gene models | Xenbase; https://www.xenbase.org/xenbase/static-xenbase/ftpDatafiles.jsp | *X. laevis* v10.1 gene models | Genomic resource. v10.1 |
| Other | *X. laevis* page IDs | Xenbase; https://www.xenbase.org/xenbase/static-xenbase/ftpDatafiles.jsp | *X. laevis* v7.1 page IDs | Genomic resource. v7.1 |
| Other | CUT&RUN for histone marks, ATAC-seq, RNA-seq in *X. laevis* | This study | GEO #GSE207027 | High-throughput sequencing data |
| Other | mir-427 gene model | miRBase; *Kozomara et al., 2019* | MI0001449 and MI0038331 | High-throughput sequencing data |
| Other | *X. laevis* wildtype stage 5 RNA-seq | *Phelps et al., 2021* | GEO #GSE152902 | High-throughput sequencing data. SRR12758941; SRR12758940 |
| Other | *X. tropicalis* wildtype RNA-seq | *Owens et al., 2016* | GEO #GSE65785 | High-throughput sequencing data. SRR1795666; SRR1795634 |
| Other | *X. tropicalis* morpholino and amanitin affected genes | *Gentsch et al., 2019* | GEO #GSE113186 | High-throughput sequencing data |

*Appendix 1 Continued on next page*

*Appendix 1 Continued*

| Reagent type (species) or resource | Designation | Source or reference | Identifiers | Additional information |
|---|---|---|---|---|
| Other | Zebrafish Pou/Sox/Nanog affected genes | *Lee et al., 2013* | GEO #GSE47558 | High-throughput sequencing data |
| Other | *X. tropicalis* acetylated enhancers | *Gupta et al., 2014* | GEO #GSE56000 | High-throughput sequencing data |
| Other | Zebrafish acetylated enhancers | *Bogdanovic et al., 2012* | GEO #GSM915197 | High-throughput sequencing data |
| Other | Chains from *X. laevis* v2 to *X. tropicalis* v9 | UCSC genome browser | xenLae2ToXenTro9 | Genomic resource for liftover. 10% minimum sequence matching |
| Other | Chains from *X. tropicalis* v2 to *X. tropicalis* v7 | UCSC genome browser | xenTro2ToXenTro7 | Genomic resource for liftover. 90% minimum sequence matching |
| Other | Chains from *X. tropicalis* v7 to *X. tropicalis* v9 | UCSC genome browser | xenTro7ToXenTro9 | Genomic resource for liftover. 90% minimum sequence matching |
| Other | Chains from *X. tropicalis* v7 to zebrafish v10 | UCSC genome browser | xenTro7ToDanRer10 | Genomic resource for liftover. 10% minimum sequence matching |
| Other | Chains from zebrafish v10 to zebrafish v11 | UCSC genome browser | danRer10ToDanRer11 | Genomic resource for liftover. 90% minimum sequence matching |

