## [Editor Report]

This paper reports fundamental findings that substantially advance our understanding of a major research question – how hybridization events influence gene regulatory programs and how evolutionary pressures have shaped these programs in response to such events. This convincing work uses appropriate and validated methodology in line with the current state-of-the-art.

---

## [Decision Letter]

**Decision letter after peer review:**

Thank you for submitting your article "Hybridization led to a rewired pluripotency network in the allotetraploid *Xenopus laevis*" for consideration by *eLife*. Your article has been reviewed by 2 peer reviewers, one of whom is a member of our Board of Reviewing Editors, and the evaluation has been overseen by Marianne Bronner as the Senior Editor. The reviewers have opted to remain anonymous.

Essential revisions:

1. Some important experiments (L624 and L672) were performed in duplicate rather than triplicate which is the standard in the field. The interpretation of extensive bioinformatic and statistical analyses is significantly weakened by opting to perform only in duplicate. These experiments need a third replicate prior to publication.

2. oct25/pou5f3.2 is also maternally expressed (but not pou5f3.1/oct91). The RNA-Seq experiment should be repeated with double MO-KD of oct60/25. As well as KD with oct25/oct60/sox3 MO.

3. L244-245-recent evidence points to an association between oct4/pou5 and *Sox2*/3 in regulating pluripotency. It would be helpful here to determine if these binding sites are oct4-*Sox2* dimer motifs.

4. The authors should dig deeper into the transcriptomics analyses in Figure 2 (and its extension to Figures 3 and 4). They identify 4700+ genes that are induced from Stage 5 to Stage 9 and more than half of these are still expressed even when CHX is added at Stage 8. They focus on this subset of 2662 genes for most of their study. Why not focus on the larger list of 4700+ induced genes? Are there interesting differences in Stage 9 L/S specific expression of maternal-zygotic vs. zygotic-only or germ layer genes? Should ideally also extend the analysis of L/S specificity of gene expression earlier to Stage 8 given that those datasets are in hand. While the authors do discriminate MZ vs. Z for some subfigures, they are missing an opportunity to dig deeper into their data. Examples include: Figure 3B: are MZ and Z differentially regulated by the 'activity' of the promoters of differentially expressed L/S genes? In Figure 4A – it appears that the authors only focused on strictly zygotic genes. Finally, what about stratifying the data into early vs. late expressed genes, or even specific germ layer genes?

5. There is some concern about the strength of evidence for differential homeolog expression or regulation linked to chromatin regulation. Need examples with a strong correlation between L/S-specific expression and L/S-specific enrichment of chromatin marks in the TSS regions using existing CUT&RUN analysis for H3K4me3 and H3K27ac. For Figure 3B, present the data as a 2D correlation plot between two parameters: the L/S ratio of chromatin marks and the L/S ratio of expression. This should reveal whether most of the genes show a correlation between activated chromatin in promoter and differential expression of homeolog. Second, even if most have no or weak correlation, the authors could focus on dozens of genes that show the strongest correlation.

6. The paper lacks direct evidence to support claims about how pluripotency factors direct specific subgenome expression at ZGA. The authors only show a general loss of gene expression from MO knockdown experiments. I expected authors would link the knockdown of Pou5f3.3 and Sox3 via MO to the loss of DIFFERENTIAL L/S expression. Is this not a likely hypothesis? Seems likely some genes would have evolved distinct regulatory control such that Pou and Sox can control differential blastula expression of one homeolog, and that their knockdown would lower expression of just the one homeolog, thus reducing L/S expression differences. The authors should provide a few examples.

7. For Figure 5, can the authors emphasize what is different or changes in the tetraploid *X. laevis*? For example, 11% of ZGA genes (400+) are expressed only from the S subgenome and thus different that X.tropicalis (which largely mirrors the L subgenome). Are these sensitive to Pou/Sox MO knockdown to a different extent than the broader set of 2662 genes characterized in this study?

---

## [Author Response]

Essential revisions:1. Some important experiments (L624 and L672) were performed in duplicate rather than triplicate which is the standard in the field. The interpretation of extensive bioinformatic and statistical analyses is significantly weakened by opting to perform only in duplicate. These experiments need a third replicate prior to publication.

We have added replicates such that our key experiments each have 3+ replicates (CUT&RUN for s8 H3K27ac, s9, H4K4me3, Pou5f3, Sox3; and ATAC-seq; also updated morpholino RNA-seq experiments; see Supp FiguresS4-8). We have updated the analyses to take these new samples into account, and have additionally converted all of the chromatin analyses to use the latest v10.1 genome assembly.

2. oct25/pou5f3.2 is also maternally expressed (but not pou5f3.1/oct91). The RNA-Seq experiment should be repeated with double MO-KD of oct60/25. As well as KD with oct25/oct60/sox3 MO.

We have now included pou5f3.2 along with pou5f3.3 and sox3 MO in a new series of combinatorial knockdowns with and without cycloheximide. MO knockdown without cycloheximide exhibit extensive downstream expression defects, with zygotic genes both up and down, which we conclude are likely due to activation of zygotic transcriptional regulators that influence secondary activation of other zygotic genes. Combining MO knockdown with CHX treatment exhibits primarily downregulation, corresponding to the direct effects of Pou5f3/Sox3. See Figure 4, Supp Figure S6.

3. L244-245-recent evidence points to an association between oct4/pou5 and Sox2/3 in regulating pluripotency. It would be helpful here to determine if these binding sites are oct4-Sox2 dimer motifs.

Although a lot of the Pou5f3 and Sox3 binding appears to target the same regulatory elements, the dimer motif seems relatively infrequent, occurring in ~10% of binding sites when we impose a conservative policy of requiring enrichment of both the individual TF motifs as well as the dimer motif. It is however possible that adjacent weaker affinity sites that would be poorly bound by one TF alone would be better bound by the heterodimer, so this may be an underestimate. Supp Figure S7F.

4. The authors should dig deeper into the transcriptomics analyses in Figure 2 (and its extension to Figures 3 and 4). They identify 4700+ genes that are induced from Stage 5 to Stage 9 and more than half of these are still expressed even when CHX is added at Stage 8. They focus on this subset of 2662 genes for most of their study. Why not focus on the larger list of 4700+ induced genes? Are there interesting differences in Stage 9 L/S specific expression of maternal-zygotic vs. zygotic-only or germ layer genes? Should ideally also extend the analysis of L/S specificity of gene expression earlier to Stage 8 given that those datasets are in hand. While the authors do discriminate MZ vs. Z for some subfigures, they are missing an opportunity to dig deeper into their data. Examples include: Figure 3B: are MZ and Z differentially regulated by the 'activity' of the promoters of differentially expressed L/S genes? In Figure 4A – it appears that the authors only focused on strictly zygotic genes. Finally, what about stratifying the data into early vs. late expressed genes, or even specific germ layer genes?

We originally reasoned the distinction between primary activation (revealed by CHX treatment) and downstream activation would be important for understanding direct versus indirect activation by maternal factors. With our updated LOF data, this has indeed turned out to be the case (Figure 4, Supp Figure S6). However, we acknowledge that for many of the properties we assayed, this distinction may not be relevant, so we heeded the advice and expanded most of the analyses to simply include all activated genes. Some distinctions between primary and secondary activated genes remain where relevant.

We have stratified the L vs S activation analyses to take into account activation timing, strictly zygotic vs maternal-zygotic genes and genes that are subsequently lineage-specifically activated (Figure 2, Supp Figure S2) and have gained additional insight regarding differential homeolog activation. There is indeed a strong homeolog-specific pattern at stage 8 that settles into a more balanced activation pattern at stage 9 (Figure 2A, Supp Figure 2A-B), which we reason is the result of timing differences in zygotic gene activation / detection of gene activation. The maternal-zygotic vs strictly zygotic analyses reveal only a slight tendency for MZ activated genes to be more homeolog-specific (Figure 2D, Supp Figure S2D) but a surprising level of homeolog choice switches between the maternal contribution and zygotic activation (Figure 2E). Lineage-specific activation also seems to be more homeolog-specific, and dissection of blastula activation reveals that the blastula both-homeolog activated genes may result from quantifying whole-embryo expression that may mix regionalized homeolog-specific activation (Figure 2D, F, Supp Figure SE-I).

Regarding chromatin, we found no significant difference in the subgenome-specific effects observed for maternal-zygotic versus strictly zygotic genes (Supp Figure S4E).

In former Figure 4A (now Figure 4B), we showed strictly-zygotic genes only for simplicity of interpretation of the biplot. We've revised the figure (Figure 4B; also Supp Figure S6D-F) to include maternal-zygotic genes, and this is accompanied by heatmap representations of expression changes (Figure 4A, Supp Figure S6B).

5. There is some concern about the strength of evidence for differential homeolog expression or regulation linked to chromatin regulation. Need examples with a strong correlation between L/S-specific expression and L/S-specific enrichment of chromatin marks in the TSS regions using existing CUT&RUN analysis for H3K4me3 and H3K27ac. For Figure 3B, present the data as a 2D correlation plot between two parameters: the L/S ratio of chromatin marks and the L/S ratio of expression. This should reveal whether most of the genes show a correlation between activated chromatin in promoter and differential expression of homeolog. Second, even if most have no or weak correlation, the authors could focus on dozens of genes that show the strongest correlation.

We've added the biplots to show the correlation between histone marks and expression differences (Supp Figure S4F), which all show significant correlations. The correlations are certainly not "perfect," and this partially underlies why we chose to focus on distal regulatory elements rather than promoters.

6. The paper lacks direct evidence to support claims about how pluripotency factors direct specific subgenome expression at ZGA. The authors only show a general loss of gene expression from MO knockdown experiments. I expected authors would link the knockdown of Pou5f3.3 and Sox3 via MO to the loss of DIFFERENTIAL L/S expression. Is this not a likely hypothesis? Seems likely some genes would have evolved distinct regulatory control such that Pou and Sox can control differential blastula expression of one homeolog, and that their knockdown would lower expression of just the one homeolog, thus reducing L/S expression differences. The authors should provide a few examples.

We think that direct regulation of activation levels through different strengths and numbers of regulatory elements between homeologs, which differentially engage pluripotency factors, is indeed a mechanism for specific subgenome expression. If LOF of the pluripotency factors leads to strongly reduced expression for both homeologs, we don't think that means a lack of support for the involvement of those factors in differential regulation.

However, the scenario posed is indeed likely, and we have expanded our MO analyses to illustrate this. Because Pou5f3/Sox3 are not the only important maternal regulators, and also seem to immediately activate zygotic factors that secondarily regulate ZGA, the regulatory networks are complex, so we can indeed see homeologs that retain activation in Pou5f3/Sox3 LOF but lose differential activation (Figure 4D-G).

7. For Figure 5, can the authors emphasize what is different or changes in the tetraploid *X. laevis*? For example, 11% of ZGA genes (400+) are expressed only from the S subgenome and thus different that X.tropicalis (which largely mirrors the L subgenome). Are these sensitive to Pou/Sox MO knockdown to a different extent than the broader set of 2662 genes characterized in this study?

There seem to be equivalently extensive regulatory differences between tropicalis and both the L and S subgenomes (Figure 5C, Supp Figure S8D-E), which are enriched among differentially expressed homeologous genes and subgenome-specific predicted regulatory elements over the shared ones. Similarly, Pou5f3/Sox3 LOF seems to affect a smaller proportion of singleton genes than genes encoded on both subgenomes, but there is not a strong L vs S bias (Supp Figure S6M). So, it's not necessarily that L more closely resembles tropicalis than S does. Rather, there seems to be a trend where higher similarity between subgenomes is correlated with higher conservation with tropicalis.